# Latent Point Collapse Induces an Information Bottleneck in Deep Neural Network Classifiers

## Abstract

The information-bottleneck principle suggests that the foundation of learning lies in the ability to create compact representations. In machine learning, this goal can be formulated as a Lagrangian optimization problem, where the mutual information between the input and latent representations must be minimized without compromising the correctness of the model's predictions. Unfortunately, mutual information is difficult to compute in deterministic deep neural network classifiers, which greatly limits the application of this approach to challenging scenarios. In this paper, we tackle this problem from a different perspective that does not involve direct computation of the mutual information. We develop a method that induces the collapse of latent representations belonging to the same class into a single point. This point collapse not only significantly reduces the entropy of the latent distribution, thereby creating an information bottleneck that correlates with improved generalization, but also makes the network Lipschitz, offering guarantees for enhanced robustness. Our method is straightforward to implement. We demonstrate that it substantially improves the network's robustness, provides a small yet statistically significant increase in generalization, and enhances the network's ability to detect misclassifications.

## 1 Introduction

Information-Bottleneck (IB) theory provides a theoretical framework for representation learning in deep neural networks (DNNs) Tishby et al. (2000); Tishby & Zaslavsky (2015). The underlying idea of IB is that, to promote more generalizable and robust learning, DNNs must discard irrelevant information. This concept is formalized by minimizing the mutual information between the input and latent representations while still retaining the information necessary for accurate prediction. This principle leads to a Lagrangian optimization problem, which has been shown to enhance the network's generalization and robustness Alemi et al. (2019). The connection between generalization and robustness is in fact well-established in the literature Xu & Mannor (2010); Szegedy et al. (2014); Achille & Soatto (2018); Novak et al. (2018). Furthermore, empirical evidence supports that IB improves network performance Hu et al. (2024), and recent theoretical work provides rigorous arguments for IB's role in controlling generalization errors Kawaguchi et al. (2023).

The IB principle suggests that DNNs aim to transform high-dimensional input data into more compact yet sufficiently informative latent representations. In fact, this transformation is automatically enforced within DNNs, as demonstrated in Shwartz-Ziv & Tishby (2017). Their findings primarily identify two dynamic phases: empirical error minimization (ERM) and compression. During the ERM phase, which occurs over a limited number of epochs, the mutual information between the layers and the input/output increases; this phase is also referred to as the fitting phase. This is followed by the compression phase, which extends over a much greater number of epochs. In this phase, the mutual information between the layers and the input decreases. During the compression phase, the network progressively constructs more compact embeddings of the input data layer by layer. In DNN classifiers, this process can be expressed as a data separation law, which demonstrates that each layer improves the separation of different classes at a constant geometric rate He & Su (2023). Such evolving separation results in the fascinating phenomenon of *Neural Collapse* (NC) Papyan et al. (2020); Han et al. (2022), which can be observed in the latent space of DNN classifiers. NC

spontaneously occurs in the penultimate layer of overparameterized DNNs during the terminal phase of training (TPT), i.e., when training continues beyond the point of zero training error. Interestingly, it has been documented that during the TPT, the network improves its generalization and robustness, implicitly suggesting a connection with the IB principle. During this terminal phase, the network continues to find more compact representations by increasing the relative distance between latent representations belonging to different classes, eventually developing a highly symmetric structure. In practice, this entails that the class means in the penultimate layer collapse to the vertices of an equiangular tight-frame simplex (ETFS).

The occurrence of NC has been widely investigated theoretically in the context of unconstrained feature models, that is when the topmost layers of the classifiers are treated as free optimization variables Mixon et al. (2020); Fang et al. (2021); Zhu et al. (2021); Graf et al. (2021); Ji et al. (2022); Tirer & Bruna (2022); Ergen & Pilanci (2020); Zhou et al. (2022a). In these idealized models, the ETFS are shown to be the only global minimizers. Nonetheless, in practical applications, perfect convergence to an ETFS is not always observed, as highlighted in Tirer et al. (2023), indicating a discrepancy between theoretical models and real-world neural networks. The phenomenon of NC has also been recently explored using information-theoretic metrics such as matrix mutual information ratio and matrix entropy difference ratio Zhang et al. (2024); Song et al. (2024b;a).

Besides improved generalization and robustness, subsequent research has revealed additional benefits of NC. For instance, in Galanti et al. (2021; 2022); Li et al. (2024) NC has been linked to transfer learning. In these works it is shown that the NC properties also emerge for classes not seen during training, provided these come from the same distribution as the training set. This implies that only a few examples are needed to train a linear classifier on these new classes. Following this trajectory, other works Wang et al. (2023); Yang et al. (2023) utilize NC-based metrics to enhance the transferability of models. Another application has been in connecting NC with out-of-distribution (OOD) detection. Haas et al. (2023) show that the appearance of NC facilitates OOD detection and subsequently Ammar et al. (2023) developed a method that leverages the NC geometric properties to enhance OOD detection.

## 1.1 Contributions

In this paper, we present a method to induce the formation of an information bottleneck in DNN classifiers. Rather than directly solving the IB Lagrangian optimization problem, we achieve this by inducing a collapse of all same-class latent representations into a single point, thereby reducing the entropy of the latent representations. This collapse is accomplished through the implementation of a penultimate layer with the following properties:

1. The layer is linear.
2. A loss function that compresses the layer's latent representations is added.
3. The dimensionality of the layer is kept low to create an information bottleneck.

A penultimate layer incorporating the three mentioned properties is referred here as *IB layer*.

An IB layer creates an information bottleneck and induces a type of collapse that substantially differs from the traditional concept of NC, which emerges spontaneously as observed in Papyan et al. (2020). Using our method, data points belonging to the same class precisely converge to a single point, and each class-specific point is located at a different vertex of a hypercube, thus developing a binary structure. This differs from NC, where such single-point convergence does not manifest, and convergence occurs toward an ETFS.

To our knowledge, this represents the first technique that forces the collapse of all latent representations of a given class into a single point, significantly enhancing class separability in the latent space. The point collapse induced by our method ensures that the network satisfies a Lipschitz condition while significantly reducing the entropy of the latent distributions, thereby enforcing an information bottleneck. Our method has practical applications, combining the benefits of Lipschitz continuity and the information bottleneck, such as improved robustness and generalization. Additionally, as demonstrated by our experiments, it enhances the network's capability to detect misclassifications.

The primary innovation of this work lies in the use of a loss function to compress the latent representations in the penultimate layer, specifically applied to a linear layer. Notably, a penalization on latent

features was already employed in the unconstrained feature model in Zhu et al. (2021) to analyze the global optimization landscape of the cross-entropy loss function, which also included penalties on both the weights and features of the layer-peeled model. However, in that theoretical study the feature penalty was used as a condition to derive a global optimizer and does not address the network enhancements that arise only under a *significant penalty* applied to a *linear layer*. We emphasize that the use of a linear layer is essential for applying a significant penalty; without it, all representations would collapse to zero, rendering them indistinguishable. In contrast, the linear layer enables the formation of binary encoding and latent point collapse—a phenomenon documented and explained with principled arguments for the first time in this work. We also demonstrate the utilization of a bottleneck layer as a penultimate layer, which has a lower dimensionality than the final classifier and the previous layer.

## 1.2 RELATED WORKS

The idea of utilizing an IB optimization in DNNs was first outlined in Tishby & Zaslavsky (2015). In the context of DNN classification, this concept has been implemented using variational approximations Alemi et al. (2019); Kolchinsky et al. (2019); Chalk et al. (2016), through injection of multiplicative noise Achille & Soatto (2017), and with efficient mutual information estimators Belghazi et al. (2021); Butakov et al. (2024); Gabrié et al. (2019); Goldfeld et al. (2019). However, minimization of the IB functional in deterministic DNNs remains ill-posed for most optimization problems Amjad & Geiger (2020). In contrast, we do not aim to minimize mutual information directly; instead, we induce the collapse of latent representations into a single point, which in turn lowers the entropy of the latent distribution and consequently the mutual information.

Sparse coding methods in deep neural networks, such as $L_0$ regularization Louizos et al. (2018), aim to sparsify the network by removing some weights or nodes Bellec et al. (2017), or input features via $L_1$ regularization (LASSO) Lemhadri et al. (2021), thereby reducing complexity and enhancing computational efficiency. These approaches, including network pruning Han et al. (2015), focus on permanently minimizing the number of active parameters to prevent overfitting and improve generalization. A conceptually different regularization technique is dropout Srivastava et al. (2014), which temporarily disables nodes during training to avoid overspecialization, but does not result in a sparse network at inference time. In contrast, our method compresses latent representations without altering the network's architecture during either training or evaluation.

The implementation of loss functions on intermediate layers of DNNs has already been utilized in the context of deep supervision Lee et al. (2014); Li et al. (2022). However, in these cases, the loss function provides additional intermediate supervision to guide representations towards the correct solution, whereas in our case, the loss function is used to compress the volume occupied by latent representations.

In the context of the NC phenomenon, the work of Haas et al. (2023) imposes $L_2$ normalization on the latent representations, projecting them onto the surface of a hypersphere. This method has been shown to accelerate convergence to NC, but it does not induce the collapse of all latent representations into a single point. Furthermore, no performance enhancement of the trained networks was documented, aside from improved OOD detection.

## 2 METHOD

Given a labeled dataset $\{\boldsymbol{x_i}, \overline{y_i}\}$, $i = 1 \ldots N$, where $N$ is the number of data points, we address the problem of predicting the labels using a classifier. We utilize a deep neural network that produces a nonlinear mapping of the input $\boldsymbol{f}(\boldsymbol{x})$, aiming to approximate the distribution represented by the data. Deep neural networks comprise multiple layers stacked together. Each layer produces an internal latent representation. The final output of the network can be expressed as a composition of the functions represented by these layers $\boldsymbol{f}(\boldsymbol{x}) = \boldsymbol{f}^{(M)} \circ \boldsymbol{f}^{(M-1)} \circ \ldots \boldsymbol{f}^{(1)}(\boldsymbol{x})$ where $M$ denotes the number of layers in the network.

For an input vector $\boldsymbol{x}$, the process of generating the neural network's output can be divided, for the sake of this exposition, into two main steps. Firstly, the nonlinear components of the neural network transform the input into a latent representation, denoted as $\boldsymbol{h}(\boldsymbol{x})$. This representation is the output of the last hidden layer before classification. The final output is then obtained by applying a linear

classifier to this latent representation: $\boldsymbol{f}(\boldsymbol{x}) = \boldsymbol{W}\boldsymbol{h}(\boldsymbol{x}) + \boldsymbol{b}$ where $\boldsymbol{W}$ and $\boldsymbol{b}$ are the weight matrix and bias vector of the linear classifier, respectively. The predicted label $y$ is computed by applying a softmax function to the network's output. The softmax function transforms the linear classifier's output into a probability distribution over classes, indicating the likelihood that the input vector $\boldsymbol{x}$ belongs to each class. The neural network is trained by minimizing the cross-entropy loss function $\mathcal{L}_{\text{CE}}(\boldsymbol{f}(\boldsymbol{x}), \overline{y}) = -\log \frac{e^{\boldsymbol{f}_{\overline{y}}(\boldsymbol{x})}}{\sum_i e^{\boldsymbol{f}_i(\boldsymbol{x})}}$ which quantifies the discrepancy between the network's predicted probabilities and the true labels.

## 2.1 Latent Point to collapse and Information bottleneck

Our objective is to design a loss function that promotes the collapse of all same-class latent representations into a single point, thereby enforcing an information bottleneck. To achieve this, we begin by formulating the optimization of the IB Lagrangian, which aims to maximize the following objective:

$$\mathcal{L}_{IB} = I(\mathbf{z}; \overline{y}) - \beta I(\mathbf{z}; \mathbf{x}), \tag{1}$$

where $I(\mathbf{z}; \overline{y})$ denotes the mutual information between the latent representation $\mathbf{z}$ and the labels $\overline{y}$ and $I(\mathbf{z}; \mathbf{x})$ represents the mutual information between $\mathbf{z}$ and the input data $\mathbf{x}$. The parameter $\beta$ controls the trade-off between compression and predictive accuracy. In App. B, we demonstrate that minimizing this quantity in deterministic DNN classification is equivalent to minimizing the following quantity:

$$\mathcal{L}_{IB} = \mathcal{L}_{\text{CE}}(\boldsymbol{f}(\boldsymbol{x}), \overline{y}) - \beta H(\mathbf{z}), \tag{2}$$

where $H(\mathbf{z})$ is the entropy associated with the latent distribution $\mathbf{z}$. Minimizing the cross-entropy loss function is a canonical approach for optimizing DNN classifiers. What remains is to minimize the entropy $H(\mathbf{z})$, which we propose to achieve indirectly by inducing the collapse of all same-class latent representations into a single point. To this end, we introduce an additional linear layer prior to the classifier, defined as $\boldsymbol{z} = \boldsymbol{W}_{\text{H}}\boldsymbol{h}(\boldsymbol{x}) + \boldsymbol{b}_{\text{H}}$. This layer serves as the penultimate step in the network architecture, with classification being subsequently determined through another linear operation, $\boldsymbol{f}(\boldsymbol{x}) = \boldsymbol{W}\boldsymbol{z} + \boldsymbol{b}$. In addition to the cross-entropy loss applied to the network's output, we incorporate an $L_2$ loss function into the penultimate layer defined as: $\mathcal{L}_{\text{H}}(\mathbf{z}) = \|\boldsymbol{z}\|^2$, where $\|\cdot\|$ is the Eucledian norm. The resulting loss function is thus composed of two terms $\mathcal{L} = \mathcal{L}_{\text{CE}} + \gamma \mathcal{L}_{\text{H}}$, where $\gamma$ is a positive scalar. The latent point collapse emerges as a result of balancing two conflicting tendencies coming from the two components of the loss function

$$\mathcal{L} = -\log \frac{e^{(\boldsymbol{W}\boldsymbol{z}+\boldsymbol{b})_{\overline{y}}}}{\sum_i e^{(\boldsymbol{W}\boldsymbol{z}+\boldsymbol{b})_i}} + \gamma \|\boldsymbol{z}\|^2. \tag{3}$$

The squared loss function encourages latent representations to be closer to zero, whereas the cross-entropy loss necessitates that the latent representations of different classes be linearly separable in the penultimate layer. This configuration generates two opposing tensions: on the one hand, the squared loss drives representations towards each other, potentially making them numerically inseparable or even overlapping; on the other hand, the cross-entropy loss enhances separability and increases the relative distance between latent representations of different classes.

In App A we provide principled arguments that describe how the interplay between these opposing tensions induce the collapse of same-class latent representations into a single point, where all collapse points are located at the same distance from the origin. Single point collapse of latent representations ensures Lipschitz continuity of the network defined as

$$\|\boldsymbol{f}(\boldsymbol{x}_1) - \boldsymbol{f}(\boldsymbol{x}_2)\| \leq L\|\boldsymbol{x}_1 - \boldsymbol{x}_2\|. \tag{4}$$

Considering that the final output representations are obtained as a linear combination of the latent representations $\boldsymbol{z}$ which are located in the sorrounding of the origin, we can conclude that latent point collapse renders networks Lipschitz. More specifically, in Zhang et al. (2022); Li et al. (2019), it is shown that the minimum amount of perturbation required to induce misclassification in a Lipschitz network is proportional to the Lipschitz constant $L$, and inversely proportional to the margin of

$f(x)$, i.e., the distance from the decision boundary. In the limit of a latent point collapse, $L$ and the margin of $f(x)$ are both proportional to distance of the collapse points from the origin. Thus, it is possible to set a minimum amount of perturbation that is necessary to trigger a misclassification. The same reasoning cannot be done in case of networks whose penultimate latent representations are not bounded close to the origin.

As shown in App. B, minimizing $I(\mathbf{z}; \mathbf{x})$ reduces to minimizing the entropy $H(\mathbf{z})$ of the latent representations. In order to understand how our method effectively minimizes the entropy associated with the probability distributions that generate the latent representations $\mathbf{z}$, we approximate the differential entropy with a discrete Shannon entropy and take the limit for an infinitesimally small quantization.

$$H_\Delta = -\sum_i p_i \log p_i \tag{5}$$

Our method induces the collapse of all same-class latent representations into a single point. As a result, all elements of a specific class are confined to a unique bin, even in the limit of a very small bin size. In case of $K$ classes where each class contains the same number of elements, the entropy reduces to:

$$H_\Delta = -\log \frac{1}{K}. \tag{6}$$

This represents the minimum possible value of entropy that still permits discrimination among classes. If the latent representations do not collapse into a single point, the distribution will be spread across multiple bins, resulting in a higher entropy. Therefore, we conclude that adding a term to the cross-entropy loss function to constrain the latent representations within a small volume enables the solution of the IB Lagrangian optimization problem.

## 2.2 BINARY ENCODING

The collapse points are located on a hypersphere, and our experiments demonstrate that they align with the vertices of a hypercube inscribed within this hypersphere. More precisely, we find that at each node of an IB layer, latent representations can approximately assume one of two values, thereby forming a binary encoding. However, we do not provide an explanation for why these collapse points specifically correspond to the vertices of a hypercube. One possible argument is that as latent representations approach the origin, it becomes increasingly difficult for the network to accurately position all representations within such a confined space without mistakenly placing a representation at a vertex associated with a different class. A practical solution to this challenge, potentially discovered by the optimizer, is to maximize the relative distance between different collapse points *in each dimension* of the latent space. This can be achieved by arranging opposing groups of collapse points symmetrically around the origin, as illustrated in Fig. 1. We note that such a symmetric arrangement can be realized using a linear layer, which allows for the symmetric displacement of latent representations relative to the origin. However, if non-linearities such as ReLU are introduced, the symmetric configuration becomes infeasible. In this case, the compression term would drive all representations toward the origin, rendering them indistinguishable.

## 3 EXPERIMENTS

The aim of the experiments is to empirically demonstrate that our method promotes a latent point collapse on the penultimate layer of DNN classifiers and to show how this is beneficial for the network's overall performance. To assess the impact of an IB layer, we conducted an ablation study using eight different network architectures. We began with a network that incorporates an IB layer, experimenting with three different dimensionalities for this layer: the lowest possible dimensionality, an intermediate dimensionality, and the same dimensionality as the previous layer. We also created three variations by individually removing each characteristic element of the IB layer. Additionally, we included two architectures that implement different regularization techniques. The first incorporates dropout on the penultimate layer before the classifier, while the second applies L1

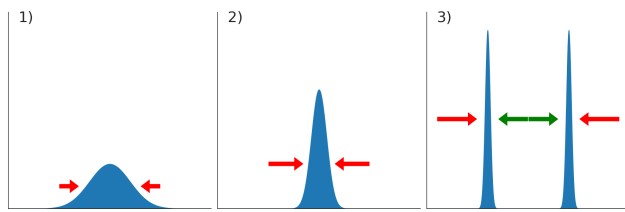

Figure 1: Graphical illustration of the dynamics leading to the emergence of a latent binary encoding. The three images give a qualitative representation of the outcome of a training where the scalar $\gamma$, as defined in Eq. equation 3, is progressively increased - from left to right - during training. Plots in the images represent histograms of the latent representations in a specific node of the linear penultimate layer. In the first image, the relatively low value of $\gamma$ constrains all values close to the origin, but the volume is still large enough for the network to differentiate between different classes in the volume. As the magnitude of $\gamma$ is increased, all latent values are drawn closer to the origin, as depicted in the second image, and it becomes increasingly more difficult for the network to discriminate between elements of different classes. Consequently, the network is forced to find, through numerical optimization, a more stable solution by placing all elements belonging to the same class in the neighborhood of one of two points. In the two distributions shown in the third figure, we observe that each of the two peaks contains elements from different classes, but all elements belonging to a specific class are confined to only one of the peaks. Through numerical optimization, these two peaks ultimately converge to a single point. This binary differentiation enables the distinction between different classes while achieving the minimum possible entropy associated with the distribution, that is what creates an information bottleneck. These points are positioned opposite to each other with respect to the origin, as illustrated in the third image, which is indeed possible with the utilization of a linear layer. The red (green) arrow represents the net effects of the binary encoding (cross-entropy) loss.

regularization to the weights of the penultimate non-linear layer with intermediate dimensionality. The use of the L1 regularizer is intended to promote sparsity in the penultimate layer, aiming to emulate the sparsification effects of our method on this layer. All architectures share a common backbone that generates the latent representation $\boldsymbol{h}(\boldsymbol{x})$, but differ in their subsequent processes for final classification. All architectures were tested on the same datasets.

The architecture with the IB layer of intermediate dimensionality is referred to as IB, while the model with the lowest dimensionality is referred to as NARROWIB, and the model with the same dimensionality as the previous layer—thus, no dimensional bottleneck—is referred to as WIDEIB. The linear penultimate (LINPEN) architecture features a linear penultimate layer with the same dimensionality as the intermediate IB layer, but is trained using only the cross-entropy loss function. The non-linear penultimate (NONLINPEN) architecture implements a non-linear penultimate layer with the same dimensionality as the other penultimate layers. The no penultimate (NOPEN) architecture performs linear classification directly on the $\boldsymbol{h}(\boldsymbol{x})$ latent representation. The (NOPENDROPOUT) architecture implements dropout on the penultimate layer of the baseline, while (NONLINPENL1) applies L1 regularization on the penultimate linear layer, that is implemented with an intermediate dimensionality.

We note that the IB, LINPEN, NONLINPEN, and NONLINPENL1 architectures have the same number of layers and parameters but differ in activation and loss functions. The WIDEIB and NARROWIB architectures have wide and narrow penultimate layers, respectively. The NOPEN and NOPENDROPOUT architectures have one fewer layer compared to the others. Only the IB, NARROWIB and WIDEIB architectures add a $\mathcal{L}_H$ quadratic loss to the cross-entropy loss $\mathcal{L}_{CE}$ as in Eq. 3, while the others are trained exclusively with the cross entropy loss $\mathcal{L}_{CE}$.

The only architecture among all possible combinations that was not considered in our study is one that implements a non-linear penultimate layer with a quadratic loss function. In Zhu et al. (2021), a light penalization on latent features was employed in their unconstrained feature model, primarily to support the theoretical derivations provided during their analysis of the global optimization landscape. In their work, they introduced a small penalty on the penultimate feature layer, which did not induce

Table 1: All values in the table represent the means and standard deviations obtained from different experiments. *Left column*: within-class covariance as defined in Eq. 7; *Central column*: coefficient of variation, defined as the ratio of the standard deviation to the mean, of the norm of the latent distributions; *Right column*: estimation of entropy of the latent distribution computed with the Kozachenko-Leonenko k-nearest neighbors method with k fixed to 100.

| DATASET: CIFAR100 | | | |
|---|---|---|---|
| MODEL | $\Sigma_W$ | COEFF. OF VAR. | ENTROPY |
| IB | $3.176 \times 10^{-12} \pm 9.552 \times 10^{-13}$ | $0.085 \pm 0.005$ | $-648.814 \pm 5.158$ |
| WIDEIB | $2.468 \times 10^{-13} \pm 4.089 \times 10^{-13}$ | $0.071 \pm 0.015$ | $-19645.291 \pm 197.479$ |
| NARROWIB | $6.599 \times 10^{-11} \pm 1.276 \times 10^{-10}$ | $0.063 \pm 0.017$ | $-85.406 \pm 3.225$ |
| NOPEN | $0.022 \pm 0.009$ | $0.102 \pm 0.071$ | $3562.544 \pm 272.490$ |
| NOPENDROPOUT | $0.017 \pm 0.009$ | $0.139 \pm 0.099$ | $3097.512 \pm 143.066$ |
| LINPEN | $0.671 \pm 0.037$ | $0.151 \pm 0.094$ | $237.878 \pm 0.385$ |
| NONLINPEN | $2.039 \pm 0.969$ | $0.176 \pm 0.097$ | $218.545 \pm 9.091$ |
| NONLINPENL1 | $0.001 \pm 0.0002$ | $0.217 \pm 0.006$ | $34.439 \pm 8.9$ |

| DATASET: CIFAR10 | | | |
|---|---|---|---|
| MODEL | $\Sigma_W$ | COEFF. OF VAR. | ENTROPY |
| IB | $1.234 \times 10^{-12} \pm 4.811 \times 10^{-13}$ | $0.033 \pm 0.009$ | $-87.598 \pm 1.706$ |
| WIDEIB | $7.890 \times 10^{-15} \pm 4.946 \times 10^{-15}$ | $0.033 \pm 0.008$ | $-5970.017 \pm 99.443$ |
| NARROWIB | $1.564 \times 10^{-12} \pm 4.882 \times 10^{-13}$ | $0.045 \pm 0.018$ | $-43.731 \pm 1.192$ |
| NOPEN | $0.029 \pm 0.005$ | $0.120 \pm 0.138$ | $467.171 \pm 69.201$ |
| NOPENDROPOUT | $0.016 \pm 0.009$ | $0.158 \pm 0.146$ | $391.265 \pm 41.978$ |
| LINPEN | $6.909 \pm 1.361$ | $0.176 \pm 0.140$ | $29.177 \pm 1.034$ |
| NONLINPEN | $17.825 \pm 7.792$ | $0.208 \pm 0.138$ | $26.799 \pm 3.044$ |
| NONLINPENL1 | $0.003 \pm 7.792$ | $0.174 \pm 0.025$ | $-4.178 \pm 1.87$ |

| DATASET: SVHN | | | |
|---|---|---|---|
| MODEL | $\Sigma_W$ | COEFF. OF VAR. | ENTROPY |
| IB | $7.705 \times 10^{-11} \pm 1.862 \times 10^{-11}$ | $0.115 \pm 0.006$ | $-74.339 \pm 0.339$ |
| WIDEIB | $2.833 \times 10^{-13} \pm 8.077 \times 10^{-14}$ | $0.114 \pm 0.007$ | $-5124.554 \pm 72.215$ |
| NARROWIB | $9.142 \times 10^{-11} \pm 1.921 \times 10^{-11}$ | $0.117 \pm 0.009$ | $-55.332 \pm 0.329$ |
| NOPEN | $0.007 \pm 0.004$ | $0.141 \pm 0.042$ | $186.138 \pm 75.137$ |
| NOPENDROPOUT | $0.009 \pm 0.005$ | $0.162 \pm 0.057$ | $143.337 \pm 62.343$ |
| LINPEN | $3.062 \pm 0.416$ | $0.165 \pm 0.052$ | $25.620 \pm 0.712$ |
| NONLINPEN | $5.409 \pm 1.891$ | $0.185 \pm 0.061$ | $23.094 \pm 2.632$ |
| NONLINPENL1 | $0.004 \pm 0.002$ | $0.268 \pm 0.044$ | $-4.037 \pm 1.559$ |

any measurable change in network performance. This penalization was used strictly in the context of theoretical analysis, rather than to achieve latent point collapse or enforce an information bottleneck as documented in this paper. To induce the phenomenon we describe, a strong penalization is required to counteract the natural tendency of cross-entropy to increase the magnitude of latent representations. However, applying such a strong penalty to layers with non-linear activations (e.g., SiLU, ReLU, or LeakyReLU) prevents the network from converging, as demonstrated by our tests. The issue arises because all latent representations are pushed close to the origin. Without access to the negative semi-axis—since the activation functions produce outputs equal to or near zero for negative inputs—the network cannot distinguish between different classes and develop binary encodings. For this reason, we exclude this specific architecture from our ablation study.

Experiments were performed on the SVHN Netzer et al. (2011), CIFAR10 and CIFAR100 datasets Krizhevsky et al. (2009) utilizing the ResNet architectures He et al. (2016). Code to reproduce the results presented in this work is available online in the linked repository [1]. All experimental details, alongside a summary with the different architectures employed, are provided in Appendix C. We remark that $L_2$ regularization was always used during training for all architectures.

---

[1]https://anonymous.4open.science/r/latent_point_collapse-A0B4

### 3.1 LATENT POINT COLLAPSE

In order to test whether a latent point collapse manifests in the penultimate layer, we study the within-class covariance, defined as:

$$\Sigma_W = \frac{1}{NP} \sum_{i=0}^{N-1} \sum_{p=0}^{P-1} \left( \boldsymbol{z}^{(i,p)} - \boldsymbol{\mu}^{(p)} \right) \left( \boldsymbol{z}^{(i,p)} - \boldsymbol{\mu}^{(p)} \right)^\top \tag{7}$$

where $\boldsymbol{z}^{(i,p)}$ is the $i$-th latent representation with label $p$, and $\boldsymbol{\mu}^{(p)}$ is the mean of all representations with label $p$. We also test whether all latent point collapses are located at the same distance from the origin, as discussed in Sec. A. Additionally, we estimate the entropy of the latent distribution on the penultimate layer using the Kozachenko-Leonenko k-nearest neighbors method.

In Table 1, we present the values of the quantities defined above at the final epoch of training for the different architectural models considered. Notably, the value of $\Sigma_W$ converges to zero only for architectures implementing an IB layer, indicating that all same-class latent representations collapse to a single point. Furthermore, we analyze the coefficient of variation for the distance of the collapse points from the origin, and observe that the collapse points consistently remains at an approximately constant distance from the origin. Lastly, we notice a sharp decrease in the entropy of the distribution due to the point collapse, which constrains all points into a small volume.

In our experiments, we demonstrate that the collapse points are located at the vertices of a hypercube, as discussed in Appendix D. Additionally, in Appendix E, we compare the latent single point collapse with the NC phenomenon. In particular, we note that our trainings were largely performed in the TPT, thus ensuring that network enhancements that we document are in addition to the ones typical of NC.

### 3.2 ROBUSTNESS, GENERALIZATION, AND RELIABILITY

In our experiments, we observe that the induction of a latent point collapse correlates with several significant benefits, including increased robustness, enhanced misclassification detection and improved generalization, as evidenced by the results presented in Table 2.

The most remarkable result is the dramatic improvement in the network's robustness. Results in Table 2 provide the magnitude of the minimal perturbation on the input data sufficient to trigger a change in the label classification. To obtain this value, we employed the DeepFool algorithm Moosavi-Dezfooli et al. (2016). Interestingly, we note that the intermediate number of nodes employed in the IB architecture provides the best results. We can explain the improved robustness with the fact that networks implementing an IB layer are Lipschitz.

The best accuracy results on the testing set at the last epoch are obtained using an IB layer, as shown in Table 2. More specifically, the best results are achieved with the architecture IB, featuring an intermediate number of nodes, that is in line with the robustness results. We can note that the addition of a penultimate layer with low dimensionality, as in the LINPEN and NONLINPEN architectures, tends to decrease the network's accuracy, and that the implementation of dropout or L1 does not provide significant improvements. This suggests that the performance improvement observed with the IB architecture is not due to an increase in the network's number of parameters nor to a regularization effect, but rather is attributed to the formation of an information bottleneck.

In order to assess the reliability of the different neural network architectures, we study the networks' confidence when making predictions. Most methods developed for assessing network predictions are focused on OOD and anomaly detection. Recently, several methods have been developed that utilize the network's latent representations Lee et al. (2018); Sastry & Oore (2019); Sun et al. (2022); Ammar et al. (2023) for OOD and anomaly detection. These methods are shown to be competitive and even the state of the art in the field Yang et al. (2022). The basic assumption is that the latent representations, specifically those of the penultimate layer, are informative about the network's understanding of the input data. By taking the latent representations of well-classified points in the training set as a reference, it is possible to construct a metric to assess the distance from those reference points. Based on this metric, we can evaluate the network's confidence. The farther a point is from other points with the same label, the less confident the network is considered to be. We perform tests employing the Mahalanobis distance method Lee et al. (2018), developed for OOD detection, but also used for

Table 2: All values in the table represent the means and standard deviations obtained from different experiments. *Left column*: Classification accuracy on the testing set at the last epoch. *Central column*: Robustness of the network computed as the norm of the minimal amount of perturbation, divided by the norm of the input, to cause a prediction change. The algorithm describing the method to produce the perturbation is in Moosavi-Dezfooli et al. (2016). We present in this table the average results across the different experiments, where in each experiment the algorithm method was tested on 100 inputs sampled from the testing set. *Right column*: AUROC values to assess reliability of the network on predictions over the entire testing set using Mahalanobis distance as score Lee et al. (2018) (left) and the ODIN method Liang et al. (2020)(right) to compute the confidence of the network. Temperature in the ODIN method was set to 100; in both methods no perturbation was added to the input.

| DATASET: CIFAR100 | | | |
|---|---|---|---|
| MODEL | ROBUSTNESS | ACCURACY | AUROC |
| IB | $\mathbf{0.247 \pm 0.011}$ | $\mathbf{77.9 \pm 0.34}$ | $0.86 \pm 0.0$ / $0.86 \pm 0.01$ |
| WIDEIB | $0.114 \pm 0.013$ | $75.87 \pm 0.15$ | $0.86 \pm 0.0$ / $\mathbf{0.87 \pm 0.0}$ |
| NARROWIB | $0.195 \pm 0.011$ | $74.9 \pm 0.47$ | $\mathbf{0.87 \pm 0.0}$ / $0.81 \pm 0.01$ |
| NOPEN | $0.008 \pm 0.0$ | $77.02 \pm 0.16$ | $0.76 \pm 0.01$ / $0.8 \pm 0.01$ |
| NOPENDROPOUT | $0.008 \pm 0.0$ | $77.13 \pm 0.09$ | $0.77 \pm 0.0$ / $0.83 \pm 0.01$ |
| LINPEN | $0.007 \pm 0.0$ | $76.51 \pm 0.24$ | $0.71 \pm 0.01$ / $0.78 \pm 0.01$ |
| NONLINPEN | $0.007 \pm 0.0$ | $76.26 \pm 0.14$ | $0.73 \pm 0.01$ / $0.76 \pm 0.01$ |
| NONLINPENL1 | $0.011 \pm 0.001$ | $76.76 \pm 0.18$ | $0.85 \pm 0.0$ / $0.86 \pm 0.0$ |

| DATASET: CIFAR10 | | | |
|---|---|---|---|
| MODEL | ROBUSTNESS | ACCURACY | AUROC |
| IB | $\mathbf{0.725 \pm 0.067}$ | $\mathbf{94.86 \pm 0.09}$ | $0.91 \pm 0.01$ / $0.84 \pm 0.01$ |
| WIDEIB | $0.336 \pm 0.119$ | $94.85 \pm 0.06$ | $\mathbf{0.92 \pm 0.01}$ / $\mathbf{0.92 \pm 0.01}$ |
| NARROWIB | $0.526 \pm 0.083$ | $94.54 \pm 0.08$ | $0.9 \pm 0.01$ / $0.77 \pm 0.02$ |
| NOPEN | $0.014 \pm 0.001$ | $94.54 \pm 0.04$ | $0.85 \pm 0.02$ / $0.69 \pm 0.03$ |
| NOPENDROPOUT | $0.016 \pm 0.001$ | $94.54 \pm 0.06$ | $0.87 \pm 0.01$ / $0.73 \pm 0.02$ |
| LINPEN | $0.014 \pm 0.0$ | $94.53 \pm 0.05$ | $0.77 \pm 0.04$ / $0.68 \pm 0.07$ |
| NONLINPEN | $0.016 \pm 0.001$ | $94.42 \pm 0.04$ | $0.88 \pm 0.01$ / $0.73 \pm 0.05$ |
| NONLINPENL1 | $0.032 \pm 0.003$ | $94.6 \pm 0.1$ | $0.91 \pm 0.0$ / $0.87 \pm 0.02$ |

| DATASET: SVHN | | | |
|---|---|---|---|
| MODEL | ROBUSTNESS | ACCURACY | AUROC |
| IB | $\mathbf{0.847 \pm 0.142}$ | $\mathbf{96.69 \pm 0.05}$ | $\mathbf{0.91 \pm 0.0}$ / $0.82 \pm 0.02$ |
| WIDEIB | $0.228 \pm 0.021$ | $96.65 \pm 0.03$ | $\mathbf{0.91 \pm 0.01}$ / $\mathbf{0.89 \pm 0.0}$ |
| NARROWIB | $0.675 \pm 0.068$ | $96.65 \pm 0.06$ | $\mathbf{0.91 \pm 0.0}$ / $0.84 \pm 0.01$ |
| NOPEN | $0.029 \pm 0.001$ | $96.4 \pm 0.03$ | $0.89 \pm 0.0$ / $0.71 \pm 0.01$ |
| NOPENDROPOUT | $0.033 \pm 0.001$ | $96.44 \pm 0.03$ | $0.9 \pm 0.0$ / $0.73 \pm 0.01$ |
| LINPEN | $0.03 \pm 0.001$ | $96.38 \pm 0.04$ | $0.88 \pm 0.0$ / $0.69 \pm 0.03$ |
| NONLINPEN | $0.032 \pm 0.001$ | $96.34 \pm 0.04$ | $0.88 \pm 0.01$ / $0.67 \pm 0.05$ |
| NONLINPENL1 | $0.06 \pm 0.004$ | $96.44 \pm 0.11$ | $0.9 \pm 0.0$ / $0.82 \pm 0.05$ |

misclassification detection Granese et al. (2021). The basic idea is that we fit a different multivariate Gaussian on each class of the train set. We then employ the mean and covariance obtained from the fit to compute the Mahalanobis distance of a new sample from the class mean. The Mahalanobis distance between a sample and the different class means can then be used to compute the probability that a sample belongs to a specific class. We compute the Mahalanobis distance from the mean of the class predicted by the network, and employ varying threshold values to accept or reject the classification prediction. In addition, we perform reliability tests using another method that employs the network output representations rather than the penultimate layer representations to assess confidence. The basic assumption is that higher values in the logits correspond to higher confidence of the network about the prediction. We employ the ODIN method Liang et al. (2020), which performs a temperature rescaling on the softmax function, with the output being the network's confidence. This confidence is then used to accept or reject predictions. The area under the receiver operating characteristic

curve (AUROC) values obtained using these two algorithms are shown in Table 2. A remarkable improvement in the network's reliability is evident when the methods are applied to a binary encoding layer. In fact, there are improvements of up to $10\%$ compared to the NOPEN architecture, which is used as a baseline for comparison. We note that for this application, a wide penultimate layer, as in the WIDEIB architecture, yields the best results.

## 4 DISCUSSION

The implementation of an IB layer assists the network in finding a compact geometry that enhances overall performance. According to the IB principle, DNNs autonomously seek progressively compact representations layer by layer to facilitate classification, eventually developing NC. However, this behavior is not consistently observed, particularly before the TPT. Following the IB principles, we design a loss function comprising two terms: one that maximizes the information about the input's label and another that encourages compression of the latent representations on the penultimate layer. The critical use of a low-dimensional linear layer enables the development of highly symmetric and compact representations, leading to the collapse of same-class latent representations into single points located at the vertices of a hypercube.

Latent point collapse renders the network Lipschitz, which proves the observed improvements in robustness. Additionally, it reduces the entropy of the latent distribution, effectively creating an information bottleneck. This phenomenon is well-documented in the literature as being associated with improved generalization, that is also evidenced in this work. These enhancements in robustness and generalization are also present in the TPT and are associated with the NC phenomenon. However, our experimental design ensures that a significant portion of the training are conducted in the TPT, ensuring that our documented benefits are supplementary to those naturally arising from NC. Beyond robustness and generalization, we also observe an improved ability of the network to detect misclassifications.

The phenomenon of single-point collapse that we have demonstrated results from the interplay between the cross-entropy loss and L2 regularization on the penultimate layer. The differing asymptotic behaviors of these two functions give rise to an equilibrium point, leading to the emergence of point collapse. As NC has been shown to develop with various types of loss functions beyond cross-entropy Han et al. (2022); Zhou et al. (2022b), it would be interesting to study this type of interplay with these other loss functions.

### 4.1 CONCLUSION

In this paper, we have introduced a straightforward method for inducing a point collapse of latent representations into the vertices of a hypercube. We have shown that this method dramatically enhances the robustness of the network, while also providing a small, but statistically significant, improvement of the generalization and reliability of the network. The simplicity of implementing this method is particularly striking, as it requires only the addition of an additional layer to an existing backbone architecture and an additional loss function. Given the ease of implementation and the significant benefits it brings to the network, we expect that this methodology will find widespread applications.

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

## A    ANALYSIS OF LATENT POINT COLLAPSE

To understand how our method induces the collapse of same-class latent representations into a single point, we analyze the behavior of the latent representations $z$ in the penultimate layer under the effect of the cross-entropy loss and the compression term.

The cross-entropy loss is defined as:

$$\mathcal{L}_{\text{CE}} = -\log \frac{e^{(\boldsymbol{W}\boldsymbol{z}+\boldsymbol{b})_{\overline{y}}}}{\sum_{i=1}^{K} e^{(\boldsymbol{W}\boldsymbol{z}+\boldsymbol{b})_i}}, \tag{8}$$

where $\boldsymbol{z} \in \mathbb{R}^d$ is the latent representation, $\boldsymbol{W} \in \mathbb{R}^{K \times d}$ is the weight matrix of the classifier, $\boldsymbol{b} \in \mathbb{R}^K$ is the bias vector, $\overline{y}$ is the true class index, and $K$ is the number of classes.

In the terminal phase of training, where all training samples are already well-classified, the latent representation $z$ is closer to the weight vector of the true label, $\boldsymbol{W}_{\overline{y}}$, than to the weight vectors of other classes. Minimizing $\mathcal{L}_{\text{CE}}$ encourages the logit corresponding to the true class, $(\boldsymbol{W}\boldsymbol{z} + \boldsymbol{b})_{\overline{y}}$, to increase relative to other logits. Since the logits are computed as a linear transformation of $z$, this process implicitly drives $z$ to adopt larger magnitudes and align more closely with the weight vector $\boldsymbol{W}_{\overline{y}}$. When $z$ is already well-aligned with $\boldsymbol{W}_{\overline{y}}$, an increase in magnitude $\|z\|$ has a larger relative effect on the projection onto $\boldsymbol{W}_{\overline{y}}$ than on projections onto other weight vectors. This is because the change in projection scales with $\cos\theta$, where $\theta$ is the angle between $z$ and the respective weight vector. As $\cos\theta$ is largest for $\boldsymbol{W}_{\overline{y}}$ in this phase, increasing $\|z\|$ further amplifies the true class logit more significantly than the logits for other classes. This results in a sharper separation of classes in the softmax probabilities and a reduction in the cross-entropy loss. It is worth noting that increasing the $\overline{y}$ element in the bias vector also contributes to softmax minimization. While this observation is accurate, it applies to all classes equally and is thus averaged out when latent representations with diverse labels are present in the same batch during optimization. Our primary focus in this study is on analyzing the alignment of latent representations with the prototype vectors of the linear classifier $\boldsymbol{W}$. Furthermore, the simplification in our analysis, where we focus solely on the behavior of the numerator, is justified by the assumption that we are already in the terminal phase of training. At this stage, latent representations are well-aligned with their respective class prototype vectors. Since the summation in the denominator involves exponentials, where one term (corresponding to the dominant class) significantly outweighs the others, changes in the total loss can be approximated by considering only the dominant term.

Geometrically, the term $\boldsymbol{W}_{\overline{y}}^{\top}\boldsymbol{z}$ can be expressed as:

$$\boldsymbol{W}_{\overline{y}}^{\top}\boldsymbol{z} = \|\boldsymbol{W}_{\overline{y}}\|\|\boldsymbol{z}\|\cos\theta,$$

where $\theta$ is the angle between $z$ and $\boldsymbol{W}_{\overline{y}}$. Minimizing $\mathcal{L}_{\text{CE}}$ involves increasing both $\|z\|$ and $\cos\theta$ to maximize this term. However, perfect alignment with $\boldsymbol{W}_{\overline{y}}$ ($\theta = 0$) is not strictly necessary. A small angular deviation $\Delta\theta$ can be compensated by increasing $\|z\|$, thereby maintaining or reducing the loss.

For small $\Delta\theta$, we approximate:

$$\cos(\theta + \Delta\theta) \approx \cos\theta - \Delta\theta\sin\theta. \tag{9}$$

Thus, the term $\boldsymbol{W}_{\overline{y}}^{\top}\boldsymbol{z}$ after the angular deviation becomes:

$$\boldsymbol{W}_{\overline{y}}^{\top}\boldsymbol{z} \approx \|\boldsymbol{W}_{\overline{y}}\|\|\boldsymbol{z}\|\left(\cos\theta - \Delta\theta\sin\theta\right). \tag{10}$$

In the overtraining regime, where $z$ is already well-aligned with $\boldsymbol{W}_{\overline{y}}$, we assume that the true label logit dominates the softmax behavior. Consequently, we only consider the terms corresponding to the true label in the exponential sum, as these dominate in the numerator and denominator. In our simplified analysis, we assume that any deviations due to other logits are exponentially suppressed and have negligible effects.

To maintain or reduce the cross-entropy loss, the softmax numerator (true label logit) must dominate the denominator. This requires the magnitude $\|z\|$ (denoted $r'$) to increase to offset the deviation. Let

$r_0 = \|\boldsymbol{z}\|$ before the deviation and $r'$ after. Substituting the dominant term in the softmax probability and comparing the exponential arguments, we obtain:

$$r'\|\boldsymbol{W}_{\overline{y}}\|(\cos\theta - \Delta\theta\sin\theta) \geq r_0\|\boldsymbol{W}_{\overline{y}}\|\cos\theta. \tag{11}$$

Rearranging gives:

$$r' \geq \frac{r_0\cos\theta}{\cos\theta - \Delta\theta\sin\theta}. \tag{12}$$

This condition ensures that the increase in $\|\boldsymbol{z}\|$ compensates for both the reduction in the true label logit and the increase in the incorrect class logit due to the angular deviation $\Delta\theta$. For small $\Delta\theta$, the denominator $\cos\theta - \Delta\theta\sin\theta$ remains close to $\cos\theta$, and the required increase in $r'$ is modest. This means the optimization process can reduce the cross-entropy loss by slightly increasing the magnitude $\|\boldsymbol{z}\|$ to compensate for the angular deviation $\Delta\theta$, without enforcing a strict collapse of $\boldsymbol{z}$ to the exact direction of $\boldsymbol{W}_{\overline{y}}$. Thus, small angular deviations from alignment do not prevent loss minimization, as the network can leverage the flexibility of increasing $\|\boldsymbol{z}\|$. This explains why, in the absence of additional constraints, latent representations do not necessarily collapse to the exact weight vector $\boldsymbol{W}_{\overline{y}}$.

To constrain $\boldsymbol{z}$ to a specific region, a compression term is added to the loss:

$$\mathcal{L}_{\mathrm{H}} = \gamma\|\boldsymbol{z}\|^2, \tag{13}$$

where $\gamma > 0$ is a regularization parameter. This term exerts a radially inward force on $\boldsymbol{z}$, counteracting the magnitude-increasing effect of the cross-entropy gradient. The total loss becomes:

$$\mathcal{L} = \mathcal{L}_{\mathrm{CE}} + \mathcal{L}_{\mathrm{H}}, \tag{14}$$

with the combined gradient:

$$\frac{\partial\mathcal{L}}{\partial\boldsymbol{z}} = -\boldsymbol{W}_{\overline{y}} + \sum_{i=1}^{K} p_i\boldsymbol{W}_i + 2\gamma\boldsymbol{z}, \tag{15}$$

where $p_i = \frac{e^{(\boldsymbol{W}\boldsymbol{z}+\boldsymbol{b})_i}}{\sum_{j=1}^{K} e^{(\boldsymbol{W}\boldsymbol{z}+\boldsymbol{b})_j}}$. The behavior of the total loss $\mathcal{L}$ depends on the magnitude of $\|\boldsymbol{z}\|$: on the one hand, for large values of $\|\boldsymbol{z}\|$, the compression term $\mathcal{L}_{\mathrm{H}}$ dominates, discouraging large deviations from the origin; on the other hand, for small values of $\|\boldsymbol{z}\|$, the cross-entropy loss $\mathcal{L}_{\mathrm{CE}}$ dominates, as the gradients from the logits $(\boldsymbol{W}\boldsymbol{z}+\boldsymbol{b})_i$ increase with the magnitude of $\|\boldsymbol{z}\|$, driving $\boldsymbol{z}$ outward. At equilibrium, these opposing forces balance, constrain $\boldsymbol{z}$ to lie on a hypersphere of radius $\Delta$, where:

$$\|\boldsymbol{z}\| = \Delta \quad\Longrightarrow\quad \| -\boldsymbol{W}_{\overline{y}} + \sum_{i=1}^{K} p_i\boldsymbol{W}_i\| = 2\gamma\Delta. \tag{16}$$

The constraint $\|\boldsymbol{z}_i\| = \Delta$ ensures that latent points are constrained to the hypersphere. However, the magnitude of $\gamma$ must be large enough to counteract the natural deviations that arise during stochastic optimization. Optimization in this setting is inherently stochastic and can be represented as a diffusive process that does not necessarily converge directly to the global minimum. Without sufficient inward pull from the compression term, such deviations may result in non-collapsed solutions where $\boldsymbol{z}$ increases in magnitude and deviates angularly, even if the network is well-trained.

Thus, the collapse of same-class representations into a unique point is only guaranteed when a pulling force, such as the compression term $\mathcal{L}_{\mathrm{H}}$, dominates sufficiently to overcome the angular deviation offset. Without this force, optimization may find alternative solutions involving increased magnitude and slight angular deviation.

## B INFORMATION BOTTLENECK IN DETERMINISTIC DNN CLASSIFIERS

The IB objective can be formulated as an optimization problem Tishby et al. (2000), aiming to maximize the following function:

$$\mathcal{L}_{IB} = I(\mathbf{z}; y) - \beta I(\mathbf{z}; \mathbf{x}), \tag{17}$$

where $I(\mathbf{z}; y)$ denotes the mutual information between the latent representation $\mathbf{z}$ and the labels $y$, while $I(\mathbf{z}; \mathbf{x})$ represents the mutual information between $\mathbf{z}$ and the input data $\mathbf{x}$. The parameter $\beta$ controls the trade-off between compression and predictive accuracy. Our goal is to maximize the mutual information between the latent representations and the labels, $I(\mathbf{z}; y)$. This mutual information can be expressed in terms of entropy:

$$I(\mathbf{z}; y) = H(y) - H(y|\mathbf{z}), \tag{18}$$

where $H(y)$ is the entropy of the labels and $H(y|\mathbf{z})$ is the conditional entropy of the labels given the latent representations. Since $H(y)$ is constant with respect to the model parameters (as it depends solely on the distribution of the labels), maximizing $I(\mathbf{z}; y)$ is equivalent to minimizing the conditional entropy $H(y|\mathbf{z})$:

$$\max I(\mathbf{z}; y) \quad \Leftrightarrow \quad \min H(y|\mathbf{z}). \tag{19}$$

The conditional entropy $H(y|\mathbf{z})$ can be estimated empirically using the dataset. Assuming that the data points $(\mathbf{x}^{(n)}, y^{(n)})$ are sampled from the joint distribution $p(\mathbf{x}, y)$ and that $\mathbf{z}^{(n)} = f(\mathbf{x}^{(n)})$, we approximate $H(y|\mathbf{z})$ as:

$$H(y|\mathbf{z}) \approx -\frac{1}{N} \sum_{n=1}^{N} \sum_{k=1}^{K} p(y_k|\mathbf{z}^{(n)}) \log p(y_k|\mathbf{z}^{(n)}), \tag{20}$$

where $K$ is the number of classes and $p(y_k|\mathbf{z}^{(n)})$ is the probability of label $y_k$ given latent representation $\mathbf{z}^{(n)}$. In practice, since we have the true labels $y^{(n)}$, this simplifies to:

$$H(y|\mathbf{z}) \approx -\frac{1}{N} \sum_{n=1}^{N} \log p(y^{(n)}|\mathbf{z}^{(n)}). \tag{21}$$

This expression corresponds to the cross-entropy loss commonly used in training classifiers. In a DNN classifier, the probability $p(y|\mathbf{z})$ is modeled using the softmax function applied to the output logits:

$$p(y_k|\mathbf{z}) = \frac{\exp\left((\mathbf{W}\mathbf{z} + \mathbf{b})_k\right)}{\sum_{i=1}^{K} \exp\left((\mathbf{W}\mathbf{z} + \mathbf{b})_i\right)}, \tag{22}$$

where $\mathbf{W}$ and $\mathbf{b}$ are the weights and biases of the final layer, and $(\mathbf{W}\mathbf{z} + \mathbf{b})_k$ denotes the logit corresponding to class $y_k$. By minimizing $H(y|\mathbf{z})$, we encourage the model to produce latent representations that are informative about the labels, aligning with the objective of accurate classification.

The second term in the IB objective, $I(\mathbf{z}; \mathbf{x})$, quantifies the mutual information between the latent representations and the inputs. To achieve compression, we aim to minimize this term. Expressing $I(\mathbf{z}; \mathbf{x})$ in terms of entropy:

$$I(\mathbf{z}; \mathbf{x}) = H(\mathbf{z}) - H(\mathbf{z}|\mathbf{x}). \tag{23}$$

In the case of deterministic mappings where $\mathbf{z} = f(\mathbf{x})$, the differntial conditional entropy $H(\mathbf{z}|\mathbf{x})$ is ill-defined, therefore we focus solely on minimizing $H(\mathbf{z})$ as explained in the InfoMax seminal paper Bell AJ (1995).

$$\min I(\mathbf{z}; \mathbf{x}) \quad \Leftrightarrow \quad \min H(\mathbf{z}). \tag{24}$$

Table 3: Summary of the features implemented in all architectures used in our ablation study. The *H loss* refers to the inclusion or exclusion of a compressing term, $\mathcal{L}_\text{H}$, in the loss function on the penultimate representation $z$. The *Nodes Add. Layer* feature indicates the presence of an additional layer between the backbone and the classification layer. If this layer is present, its dimensionality is categorized as one of three possible values: "wide," "intermediate," or "narrow." The exact dimensionality for these categories is a hyperparameter that varies across different datasets. *Linear* indicates whether the penultimate layer is linear, without a non-linear activation function. *Dropout* indicates whether dropout is applied to the penultimate layer before classification.

| Model | H Loss | Linear | Dropout | L1 | Nodes Add. Layer |
|---|---|---|---|---|---|
| IB | ✓ | ✓ | ✗ | ✗ | Intermediate |
| WideIB | ✓ | ✓ | ✗ | ✗ | Wide |
| NarrowIB | ✓ | ✓ | ✗ | ✗ | Narrow |
| NoPen | ✗ | ✗ | ✗ | ✗ | ✗ |
| NoPenDropout | ✗ | ✗ | ✓ | ✗ | ✗ |
| LinPen | ✗ | ✓ | ✗ | ✗ | Intermediate |
| NonlinPen | ✗ | ✗ | ✗ | ✗ | Intermediate |
| NonlinPenL1 | ✗ | ✗ | ✗ | ✓ | Intermediate |

## C  Training and Architecture Details.

To generate the latent representation $h(x)$, two distinct ResNet He et al. (2016) backbone architectures, of increasing complexity, were employed for three different datasets, also chosen of increasing complexity. A ResNet18 architecture was utilized for the SVHN Netzer et al. (2011) and CIFAR10 Krizhevsky et al. (2009) datasets; a ResNet50 architectures was utilized for CIFAR100 dataset Krizhevsky et al. (2009). The ResNet architectures implemented batch normalization, and the swish activation function Ramachandran et al. (2017) was employed in all non-linear layer. The IB, LinPen, NonlinPenL1, and NonlinPen architectures included an additional fully connected penultimate layer consisting of 8 nodes when trained on CIFAR10 and SVHN, and 64 nodes when trained on CIFAR100. The WideIB architecture implemented an additional fully connected penultimate layer, with the same dimensionality as the previous layer, and thus consisting of 512 nodes when trained on SVHH and CIFAR10, and 2048 nodes when trained on CIFAR100. The NarrowIB architecture included an additional fully connected penultimate layer consisting of 6 nodes when trained on SVHH, 4 nodes when trained on CIFAR10 and 8 nodes when trained on CIFAR100. The NoPenDropout architecture implemented dropout on the penultimate - non linear - layer, with probability of elements to be zeroed always set to $0.5$. The NonlinPenL1 architecture implemented dropout on the penultimate - non linear - layer, with penalty set to $10^{-4}$. A summary of the different architectures employed is provided in Table 3.

The training process utilized the AdamW Kingma & Ba (2017); Loshchilov & Hutter (2019) optimizer with default settings, as specified in the PyTorch implementation, and a weight decay set to $0.5 \times 10^{-4}$. Data were augmented during training using random horizontal flips, random cropping with padding set to $4$, and the addition of Gaussian noise with a fixed standard deviation of $0.01$. Each experiment consisted of conducting different training sessions with varying learning rates. The session that yielded the best value on the validation set in the last epoch was selected. The set of learning rates was chosen from a geometric sequence, starting from an initial value of $10^{-4}$. Each subsequent value was obtained by multiplying the previous one by a factor of $2$. A total of $5$ different initial learning rates were employed, spanning one order of magnitude. The learning rates are halved every 50 epochs starting from epoch 250 during training. However, the parameters of the last linear classifier are not affected by this scheduled optimization.

All training sessions were conducted for 800 epochs, and performance metrics were recorded every 50 epochs. We opted for extended simulations to ensure that the majority of the training occurs during the terminal phase of training (TPT). The TPT is defined as the period when the network has already achieved convergence in terms of accuracy on the training set. The onset of the TPT is marked by the epoch at which the accuracy on the training set reaches 99.9% Papyan et al. (2020).

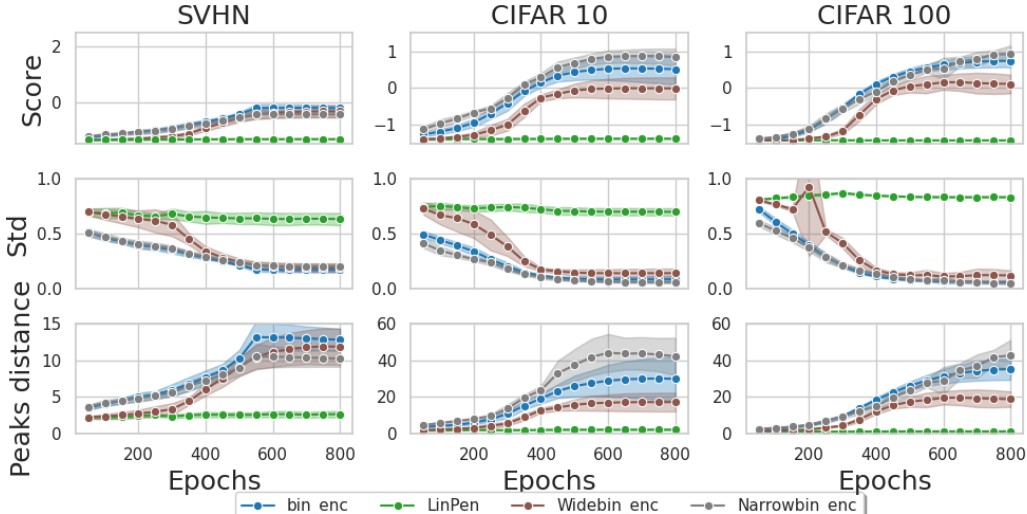

Figure 2: Log-likelihood scores, standard deviations and weighted peak distance of bimodal Gaussian mixture models fitted on each dimension of the penultimate layer using all values of the train set. From top to bottom, the quantities described as $\bar{\ell}$, $\bar{\sigma}$, and $\bar{\mu}$ in Eqs. 25, 26, 27 are computed for the IB, WIDEIB, NARROWIB, LINPEN, and LINPENDROPOUT architectures. We can see that the latent representations in the binary encoding layer, including its wide version, are well represented by two Gaussians with increasingly small standard deviations, while this is not observed for the LINPEN and LINPENDROPOUT architectures. We note that ht

The loss function employed for the IB, WIDEIB and NARROWIB architectures followed Eq. equation 3, with the value of $\gamma$ starting from a low value and increasing at each epoch. For all datasets, the initial value of $\gamma$ was set to $10^{-2}$. This value was then multiplied by a factor, $\gamma_{step}$, at each epoch, up until it reached the value of $\gamma_{max}$. Once this maximum value, $\gamma_{max}$, was reached, $\gamma$ was no longer increased. For the SVHN dataset, $\gamma_{step} = 1.03$ and $\gamma_{max} = 10^5$. For both the CIFAR-10 and CIFAR-100 datasets, $\gamma_{step} = 1.05$ and $\gamma_{max} = 10^6$.

For each network architecture and dataset, 5 independent experiments were performed. Each model is trained independently from a different random weight initialization in every experiment. The quantities displayed in the plots represent the averages and standard deviations of these outcomes.

## D  BINARITY HYPOTHESIS

Our assumption is that each dimension on the penultimate latent representation can assume approximately only one of two values. In order to verify this assumption, we fit a Gaussian mixture model (GMM) with 2 modes on each set of latent representations $z_i \sim \mathcal{N}\left(\mu_i^{(1,2)}; \sigma_i^{(1,2)^2}\right)$. For each dimension $i$, we build a histogram with the values of all latent representations of the train set. We then fit a bimodal GMM model on this histogram. Assuming that $P$ is the dimensionality of the latent representation and the dataset contains $N$ datapoints, the following quantities are collected: The average log-likelihood score

$$\bar{\ell} = \frac{1}{NP} \sum_{n=0}^{N-1} \sum_{i=0}^{P-1} \log \mathcal{N}\left(z_i^{(n)} \middle| \mu_i^{(1,2)}; \sigma_i^{(1,2)^2}\right); \tag{25}$$

Table 4: Score $\overline{\ell}$ and relative distance of the two distances $\overline{\mu}$ over all penultimate nodes in the training set across all experiments at the last epoch. Average and min values are shown. The coefficient of variation measures the standard deviation of the norm of latent representations normalized by the mean.

| | | DATASET: CIFAR100 | | | |
|---|---|---|---|---|---|
| MODEL | SCORE | MIN SCORE | PEAKS DIST | MIN PEAKS DIST | COEFF. OF VAR. |
| IB | $0.745 \pm 0.194$ | 0.358 | $35.262 \pm 6.655$ | 23.554 | 0.004 |
| WIDEIB | $0.113 \pm 0.291$ | -1.405 | $18.878 \pm 4.867$ | 4.79 | 0.052 |
| NARROWIB | $0.931 \pm 0.251$ | 0.165 | $42.15 \pm 24.774$ | 14.189 | 0.036 |
| LINPEN | $-1.42 \pm 0.003$ | -1.423 | $1.364 \pm 0.15$ | 0.839 | 0.189 |
| | | DATASET: CIFAR10 | | | |
| MODEL | SCORE | MIN SCORE | PEAKS DIST | MIN PEAKS DIST | COEFF. OF VAR. |
| IB | $0.516 \pm 0.5$ | -0.259 | $29.955 \pm 15.769$ | 12.472 | 0.012 |
| WIDEIB | $-0.006 \pm 0.436$ | -1.102 | $17.273 \pm 8.096$ | 5.027 | 0.018 |
| NARROWIB | $0.845 \pm 0.503$ | -0.137 | $42.15 \pm 24.774$ | 14.189 | 0.023 |
| LINPEN | $-1.374 \pm 0.055$ | -1.421 | $2.267 \pm 0.698$ | 1.098 | 0.224 |
| | | DATASET: SVHN | | | |
| MODEL | SCORE | MIN SCORE | PEAKS DIST | MIN PEAKS DIST | COEFF. OF VAR. |
| IB | $-0.228 \pm 0.279$ | -0.952 | $12.783 \pm 3.618$ | 6.087 | 0.117 |
| WIDEIB | $-0.342 \pm 0.349$ | -1.349 | $11.851 \pm 3.109$ | 5.027 | 0.1 |
| NARROWIB | $-0.456 \pm 0.228$ | -0.946 | $10.226 \pm 2.401$ | 6.097 | 0.117 |
| LINPEN | $-1.349 \pm 0.053$ | -1.421 | $2.655 \pm 0.844$ | 0.11 | 0.118 |

the average standard deviation of the two posterior distributions

$$\overline{\sigma} = \frac{1}{P} \sum_{i=0}^{P} \left( \frac{\sigma_i^{(1)} + \sigma_i^{(2)}}{2} \right);$$  (26)

and the mean relative distance of the two peaks reweighted with the standard deviation

$$\overline{\mu} = \frac{1}{P} \sum_{i=0}^{P} \frac{\left\| \mu_i^{(2)} - \mu_i^{(1)} \right\|}{\left( \sigma_i^{(1)} + \sigma_i^{(2)} \right)/2}.$$  (27)

These values are plotted in Fig. 2, showing that during training, the log-likelihood score increases while the standard deviation decreases. Additionally, the relative distance between the two modes of the GMM models increases. These three metrics indicate that during training, all latent representations collapse into two distinct points, forming two clearly separated clusters. This observation supports our binarity hypothesis, which states that each latent representation can assume only one of two possible values. The same analysis was performed for the LINPEN architecture, which also features a linear layer before classification. However, in this architecture, the binarity hypothesis does not hold.

To validate our claim that binary encoding manifests in each node of the penultimate layer, we present averages in Table 4. These averages are computed across all nodes in the penultimate layer from all experiments. The table shows the average and minimum values for the GMM fitting score and the weighted relative distance between the peaks across all nodes and experiments. For all IB architectures—defined as those with a linear layer, latent compression, and a dimensional bottleneck (e.g., IB and NARROWIB)—the binarity hypothesis holds true for all dimensions, even in cases with the lowest recorded scores. Although the WIDEIB architecture shows instances where the GMM assumption does not hold true in the worst-case scenario (see the minimum values), the distance between the two peaks remains large even in these cases. This suggests that each node still forms two distinct clusters, supporting the idea that they can be grouped into binary values. The table also includes the coefficient of variation for the norm of the latent representations. These values approach

zero for all IB architectures across the CIFAR10 and CIFAR100 datasets, indicating that the latent representations are equidistant from the origin in each dimension. This observation supports our claim that the class means of the latent representations collapse onto the vertices of a hypercube. However, the coefficients of variation do not approach zero in the SVHN dataset. Nonetheless, as shown by $\Sigma_W$ in Table 1, all latent point representations of the same class in the SVHN dataset do collapse into a single point. Therefore, we conclude that for this specific dataset, the representations do not perfectly align to the vertices of a hypercube. However, it is important to note that even in the worst-case scenarios within the SVHN dataset, the peak distance remains significant. Thus, we conclude that the binarity hypothesis still holds in this case.

The emergence of binary encoding is made possible because all latent representations collapse into one of two possible points. This collapse reflects a natural tendency of DNNs to seek increasingly compact embeddings of latent representations. The phenomenon of NC can also be viewed as an expression of this inclination, though it significantly differs from the binary encoding induced by our method. More specifically, NC involves the convergence of class means to the vertices of an ETFS, but this is not possible in a low-dimensional manifold. In fact, an ETFS constructed in a $P$ dimensional space contains $P + 1$ vertices, thus if the number of classes is equal to $K$, it is not possible to assign each class to a different vertex when $P < K - 1$. In this case of large number of classes, NC converges to a structure where class means are at equidistant points on the surface of a hypersphere Jiang et al. (2024); Liu et al. (2023). In our experiments, this condition was always met. More specifically, for the CIFAR10 dataset, the embedding utilized with NARROWIB had the lowest possible dimensionality to preserve information about different classes using a binary encoding, i.e., $P = \lceil \log_2 K \rceil$. We also note that, a hypercube constructed in a $P$-dimensional space contains $2^P$ distinct vertices, with each of the $K$ different classes occupying one of those vertices. All other $2^P - K$ vertices remain unoccupied, suggesting that they could potentially be utilized to learn new classes in a transfer-learning setting.

Collapse into the vertices of a hypercube creates a latent binary encoding of the input representations onto the penultimate layer. In a classification task that includes $K$ different classes, a dimensionality of $\lceil \log_2 K \rceil$ is required in the penultimate layer to create the minimum amount of information for a binary encoding to encode the input signal. Thus, if the penultimate layer has a larger dimensionality, in a number of nodes all representations could collapse to zero to minimize the compressing term in the loss function while maintaining the same accuracy. However, this scenario is never verified in our case studies where a binary structure is observed in each node, even when a redundant penultimate layer (i.e., with a dimensionality larger than $\lceil \log_2 K \rceil$) is utilized. We could explain this with the consideration that through gradient-based optimization methods, such a dramatic change of configuration—i.e., all latent representations in one node collapsing to zero without altering classification performance of the network—cannot be reached in practice. Still, in our case studies, we note that more performant networks, in terms of generalization, implement a dimensionality of the penultimate layer that is larger than the minimum value that would be required only considering the total number of classes. In these cases some nodes are redundant for a basic encoding of the class. One possible reason for utilizing more dimensions than necessary in the penultimate layer could be that, during training, the network progressively learns to employ each dimension in the penultimate layer for signaling specific features that are extracted from the input data, thus overcoming the idea that the binary encoding is only meant to differentiate between the different classes. According to this hypothesis, the binary encoding would entail that each dimension in the penultimate layer signals the presence or absence of a specific feature that is identified by the network in the different classes. This hypothesis, if verified, would then pave the way for future applications of binary encoding for explainability in deep neural networks.

# E  NEURAL COLLAPSE

In this appendix, we present all metrics related to NC as defined in in Papyan et al. (2020). The entire NC phenomenon can be summarized into four distinct components: (1) the variability of samples within the same class diminishes as they converge to the class mean (NC1); (2) the class means in the penultimate layer tend towards an ETFS (NC2); (3) the last layer classifier weights align with the ETFS in their dual space (NC3); and (4) classification can effectively be reduced to selecting the closest class mean (NC4).

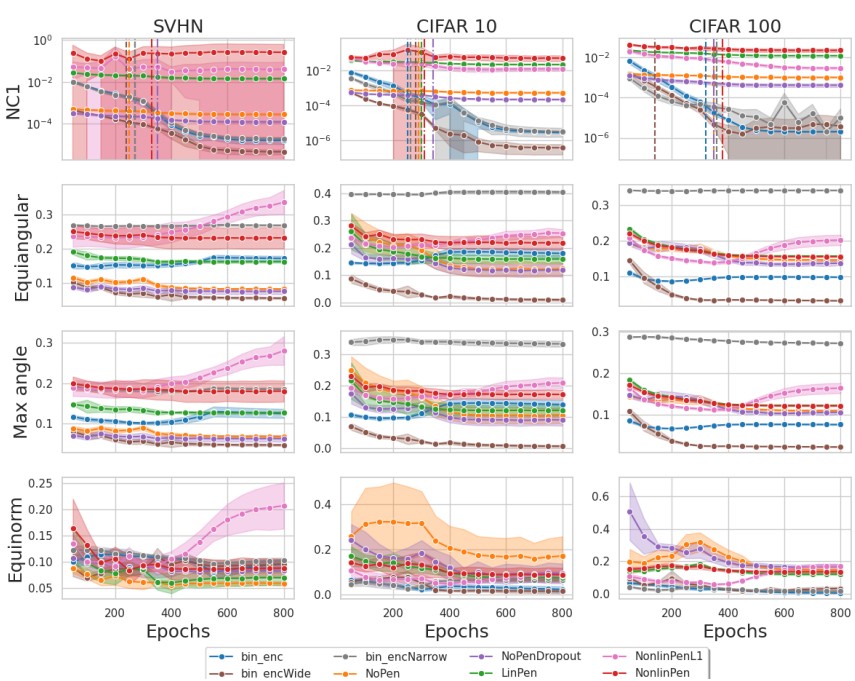

Figure 3: Metrics used to evaluate convergence towards Neural Collapse (NC). In the upper figure, we examine a renormalized version of the NC1 property. This normalization process is conducted based on the number of nodes in the penultimate layer to ensure a fair comparison across models with varying dimensions of the penultimate layer. The dashed lines are drawn at the average epoch when training reaches convergence, that demonstrates that most of the training was performed in the TPT. Below, we present metrics demonstrating convergence to an ETFS, utilizing the same parameters as those outlined in Papyan et al. (2020).

The first property of interest is NC1, which asserts that the variability of samples within the same class decreases in the terminal phase of training. This property is characterized by the equation $\mathrm{Tr}\left(\mathbf{\Sigma}_W \mathbf{\Sigma}_B^\dagger / K\right)$, where $\Sigma_W$ is defined as

$$\Sigma_W = \frac{1}{NP} \sum_{i=0}^{N-1} \sum_{p=0}^{P-1} \left(\mathbf{z}^{(i,p)} - \boldsymbol{\mu}^{(p)}\right)\left(\mathbf{z}^{(i,p)} - \boldsymbol{\mu}^{(p)}\right)^\top \tag{28}$$

where $\mathbf{z}^{(i,p)}$ is the $i$-th latent representation with label $p$ and, $\boldsymbol{\mu}^{(p)}$ is the mean of all representation with label $p$; and $\Sigma_B$ is defined as:

$$\Sigma_B = \frac{1}{P} \sum_{p=0}^{P-1} \left(\boldsymbol{\mu}^{(p)} - \boldsymbol{\mu}_G\right)\left(\boldsymbol{\mu}^{(p)} - \boldsymbol{\mu}_G\right)^\top. \tag{29}$$

The trace operation sums over all diagonal elements, the dimensionality of which is equal to that of the penultimate layer, $P$. Given the use of different architectures with varying numbers of nodes in the penultimate layers in our study, we examine a renormalized version of this quantity, $\mathrm{Tr}\left(\mathbf{\Sigma}_W \mathbf{\Sigma}_B^\dagger / K / P\right)$. In Fig 3, the top image presents this value, showing that it is at least an order of magnitude lower in the IB architecture compared to the baseline architecture. This demonstrates that

only the implementation of a binary encdoding layer ensures convergence of all representations of the same class to a unique point, even when considered relative to the distances between class means.

The other three images below demonstrate convergence of the class means towards ETFS, also known as the NC2 property. These images show that all values reach a plateau in the terminal phase, indicating convergence to their optimal values. It is evident that architectures with a dimensional bottleneck in the penultimate layer tend to exhibit higher values for convergence to the angular values (maximum angle and equiangularity) compared to the baseline. This is because geometrically, it is impossible to construct an ETFS with a number of vertices equal to the total number of classes in such a dimensional bottleneck. However, this limitation changes in the case of the CIFAR-100 dataset, where the bottleneck is less pronounced. In this case, even when dimensionality is low, a good approximation of the ETFS can still be found. Interestingly, we note that the WIDEIB architecture, which is the only architecture with a compression loss on a penultimate layer that is not a dimensional bottleneck, has the stronger convergence towards an ETFS. In fact, this is consistent with the observation that single point collapse is a stronger manifestation of NC.

