# OpenReview forum: "Latent Point Collapse Induces an Information Bottleneck in Deep Neural Network Classifiers"
_ICLR.cc/2025/Conference — Submitted to ICLR 2025_

### Official Review · Reviewer_gEVm · 2024-10-30

**Soundness:** 2
**Presentation:** 2
**Contribution:** 1
**Rating:** 5
**Confidence:** 3

**Summary:**

The paper presents an innovative approach to enhancing robustness and generalization in deep neural network classifiers. The authors introduce an additional linear layer as the latent feature layer, along with a regularization term for the learned latent features. They claim that this method creates compact representations of input data by inducing a latent point collapse, where same-class representations converge into single points in the latent space, effectively reducing entropy and creating a bottleneck effect without the need to directly compute mutual information.

**Strengths:**

The proposed method is easy to implement and slightly increases classification accuracy.

**Weaknesses:**

1. **Lacking of rigorous analysis**. Although the authors claim that their method leads to the collapse of all same-class representations into a single point, there is no theoretical proof to substantiate this claim.
2. **Lacking of experiment proof**. The experiments show that the entropy and covariance of latent features decrease, but these metrics could naturally decline due to the regularization term applied to z . Clear evidence of latent feature collapse is lacking, such as metrics comparing the distance between class centers or between the origin and class centers.

**Questions:**

Currently, the paper is purely hypothetical, lacking both theoretical and empirical evidence of latent point collapse.

---

> ### Author Response · Authors · 2024-11-19
>
> We thank the reviewer for thoroughly reading the manuscript. Please find below our responses to all points raised.
>
> **Q1**: *Lacking of rigorous analysis. Although the authors claim that their method leads to the collapse of all same-class representations into a single point, there is no theoretical proof to substantiate this claim.*
>
> **A1**: The single-point collapse arises from the interplay between cross-entropy, which promotes differentiation of points belonging to different classes and increases the magnitude of latent representations, and the compression term, which pulls latent representations toward the origin.
> In the revised version, we have included a principled explanation of this phenomenon in Appendix A ("Analysis of Latent Point Collapse").
>
> **Q2**: *Lacking of experiment proof. The experiments show that the entropy and covariance of latent features decrease, but these metrics could naturally decline due to the regularization term applied to z . Clear evidence of latent feature collapse is lacking, such as metrics comparing the distance between class centers or between the origin and class centers.*
>
> **A2**: The reviewer is absolutely correct in noting that these metrics decline due to the regularization applied to z. This is a novel contribution introduced in this paper, demonstrating that applying strong regularization to a penultimate linear layer induces latent point collapse.
> Evidence of feature collapse is provided in Table 1, where the coefficient of variation (the ratio of the standard deviation to the mean) of the norm of the latent distribution is shown. This indicates that the means of all classes are located at the same distance from the origin. Furthermore, the collapse of the within-class covariance, denoted as $\Sigma_w$, suggests that all latent representations for a single class collapse into the same point.
> However, the collapse cannot occur at the same point for different classes, as this would render the final linear classifier unable to discriminate between them. Instead, each class point must occupy a distinct position equidistant from the origin.
> What we empirically demonstrate is that these class points are located at the vertices of a hypercube, leading to the phenomenon of binary encoding. Each class is thus mapped to a distinct vertex. This is detailed in Appendix B ("Binarity Hypothesis"), where we fit a Gaussian mixture model with two modes to the latent representations along each dimension. The results show that in the case of an IB layer, this model achieves a good fit.
> Additionally, we observe that during training, the standard deviations of the two Gaussians tend to zero, further confirming point collapse. We also measure the distance between the peaks, reweighted by the standard deviation of the two Gaussians. These values are significantly large (ranging from 10 to 50), indicating that the two peaks are well-separated relative to their standard deviations.
> This demonstrates that latent representations effectively collapse into one of two distinct and well-separated points, further supporting our claims.
>
> Given the considerable effort we have devoted to thoroughly addressing the reviewers' comments during the rebuttal process, we kindly request that you reconsider your score. Your positive feedback would mean a great deal to us and is sincerely appreciated.

---

> > ### Comment · Reviewer_gEVm · 2024-11-24
> > **My concerns remain unresolved**
> >
> > Thank the authors for providing clarification. However, my concerns remain unresolved:
> >
> > 1. The analysis appears highly heuristic and lacks sufficient rigor.  For example, how do you define the weight vector of the true label $W_{\bar{y}}$? Is it the $\bar{y}$th row or column of $W$, or some optimal latent vector for $\bar{y}$ you are assuming? And why is optimizing the cross-entropy loss equivalent to maximizing $W_{\bar{y}}^T z$? Are you assuming the bias $b$ is zero? It requires rigorous mathematical deduction to claim equivalence. Even it is true, are the authors assuming $W_{\bar{y}}$ forms a hypercube? Note that projecting a hypercube from $\mathbb{R}^d$ to $\mathbb{R}^K$ does not necessarily results in a hypercube. Similarly, $W_{\bar{y}}$ does not need to form a hypercube for its low-dimensional projection to exhibit such a structure.
> > 2. The authors did not "empirically demonstrate that these class points are located at the vertices of a hypercube" as they claimed. While a small $\Sigma_w$ implies that latent representations for the same class collapse into a single point, the assertion that “each class point must occupy a distinct position equidistant from the origin” seems overly strong. I do not see any evidence suggesting that these points are equidistant from the origin. Even if they are equidistant, this would imply they lie on a hypersphere, not necessarily at the vertices of a hypercube.

---

> > > ### Author Response · Authors · 2024-11-25
> > >
> > > We thank the reviewer for their comments on our rebuttal and for highlighting some unresolved issues. Please find our responses to all the points raised below.
> > >
> > >
> > > **Q1**: *How do you define the weight vector of the true label $\boldsymbol{W}_{\overline{y}}$? Is it the ${\overline{y}}$-th row or column of $\boldsymbol{W}$, or some optimal latent vector for you are assuming?*
> > >
> > > **A1**: $\boldsymbol{W}_{\overline{y}}^\top$ represents the y-th row of the matrix $\boldsymbol{W}$ and the dot product of this vector with the latent representation $\boldsymbol{z}$ results into the $y-th$ logit, which in case of the true label ${\overline{y}}$ is the argument of the exponent of the numerator of the softmax function.
> > >
> > >
> > > **Q2**: *Why is optimizing the cross-entropy loss equivalent to maximizing $\boldsymbol{W}_{\overline{y}}^\top \boldsymbol{z}$ ? Are you assuming the bias is zero?*
> > >
> > >  **A1**: Thank you for pointing out this assumption. It has now been explicitly mentioned in the revised version of the manuscript that we just uploaded (see lines 769->787). The reviewer may have observed that increasing the ${\overline{y}}$ element in the bias vector also leads to a softmax minimization. While this observation is accurate, it applies to all classes. Consequently, this effect is averaged out when latent representations with diverse labels are present in the same batch during optimization. Our primary focus in this study is to analyze the alignment of latent representations with the prototype vectors of the linear classifier $\boldsymbol{W}$.
> > > Furthermore, the simplification in our analysis, where we focus solely on the behavior of the numerator, is justified by the assumption that we are already in the terminal phase of training. At this stage, latent representations are well-aligned with their respective class prototype vectors. Since the summation in the denominator involves exponentials, where one term (corresponding to the dominant class) significantly outweighs the others, changes in the total loss can be approximated by considering only the dominant term. This justification has also been clarified in the revised version of the manuscript.
> > >
> > > **Q1**: *Are the authors assuming $\boldsymbol{W}$ forms a hypercube?*
> > >
> > > **A1**: This is not explicitly mentioned in the manuscript, but it can be inferred from the fact that the latent representations collapse onto the vertices of a hypercube and that, in the terminal phase of training, the linear classifier aligns well with the latent representations.
> > >
> > >
> > > **Q1**: *The authors did not "empirically demonstrate that these class points are located at the vertices of a hypercube" as they claimed. I do not see any evidence suggesting that these points are equidistant from the origin. Even if they are equidistant, this would imply they lie on a hypersphere, not necessarily at the vertices of a hypercube.*
> > >
> > > **A1**: To further support our claim that these points are located on the vertices of a hypercube, we included an additional analysis in Appendix D "Binarity Hypothesis" in the revised version of the manuscript. In the "Coeff. of Var." column of Table 4, we present the coefficient of variation of the norm in each dimension of the penultimate latent representations across all five experiments we conducted. The coefficient of variation is defined as the standard deviation of a quantity divided by its mean norm. When this value approaches zero, it indicates that the target quantity remains approximately constant, weighted by the standard deviation. In our case, this implies that each latent representation is nearly equidistant from the origin in every dimension, providing further justification for our claim that latent representations lie on the vertices of a hypercube.
> > > However, we also note that in case of the SVHN dataset there is not a perfect alignment with the vertices of a hypercube, but the binarity hypothesis still holds.
> > >
> > > We kindly ask the reviewer to let us know if there are any remaining concerns. Otherwise, if all concerns have been addressed, we would appreciate it if the reviewer could consider raising their score.

---

> > > > ### Author Response · Authors · 2024-11-27
> > > > **Deadline approaching**
> > > >
> > > > Dear Reviewer,
> > > >
> > > > As the deadline to upload a revised version of the manuscript is approaching, we kindly ask if you still have any remaining concerns.
> > > >
> > > > Thank you for your attention.

---

> > > > ### Comment · Reviewer_gEVm · 2024-12-02
> > > >
> > > > Thank you to the authors for their clarification. The revised manuscript is significantly clearer. I have raised my score to 5 and the soundness score to 2, as I still feel there are some limitations in both the theoretical and empirical contributions.

---

> > > > > ### Author Response · Authors · 2024-12-03
> > > > > **Thank you**
> > > > >
> > > > > Thank you for engaging with this revision and for improving the score.

---

### Official Review · Reviewer_vLMu · 2024-10-31

**Soundness:** 2
**Presentation:** 3
**Contribution:** 2
**Rating:** 3
**Confidence:** 4

**Summary:**

The paper introduces a method that induces the collapse of latent representations belonging to the same class into a single point, creating an information bottleneck in machine learning models. By focusing on reducing the entropy associated with the latent distribution, the method enhances the network's robustness, generalizability, and reliability without the need for direct computation of mutual information.

**Strengths:**

>The paper introduces a novel method to avoid calculating mutual information in IB settings.

>Some improvements are achieved on 3 standard benchmarks.

**Weaknesses:**

>The improvements seem to be marginal, for e.g. in Table 1.

>Some more experiments are expected on larger datasets like ImageNet.

>Some theoretical benefits of applying the proposed method in IB are needed.

>There seem to be discussions of neural collapse and information theory in the literature [1] [2] that are closely related but are not discussed.

[1]: Matrix Information Theory for Self-Supervised Learning. ICML 2024

[2]: Unveiling the Dynamics of Information Interplay in Supervised Learning, ICML 2024

**Questions:**

See weaknesses.

---

> ### Author Response · Authors · 2024-11-19
>
> We thank the reviewer for thoroughly reading the manuscript. Please find below our responses to all points raised.
>
> **Q1**: *The improvements seem to be marginal, for e.g. in Table 1.*
>
> **A1**: We would like to highlight that Table 1 demonstrates how our method induces latent point collapse.
> Specifically, it shows that only architectures implementing an IB layer achieve within-class covariance values approaching zero—approximately ten orders of magnitude lower than baseline methods.
> Additionally, Table 1 illustrates that latent point collapse leads to a significant reduction in entropy, and that all same-class convergence points are approximately equidistant from the origin.
> The results we kindly ask the reviewer to evaluate are presented in Table 2. More specifically, the main achievement of our method is an over tenfold improvement in robustness. This enhancement in robustness is clearly emphasized as the primary achievement of our method in the revised version. We also note a marginal yet statistically significant improvement in generalization.
>
> **Q2**: *Some more experiments are expected on larger datasets like ImageNet.*
>
> **A2**: As stated in our general rebuttal comment, conducting experiments on large-scale datasets is beyond our current computational resources.
> The datasets we use in this study are already widely employed in many impactful works published at top machine learning conferences. Furthermore, we corroborate our experimental results with principled explanations of the phenomenon we identify.
>
> **Q3**: *Some theoretical benefits of applying the proposed method in IB are needed.*
>
> **A3**: In Appendix B ("Information Bottleneck in Deterministic DNN Classifiers"), we demonstrate that minimizing the IB Lagrangian optimization problem can be reduced to minimizing the entropy of the probability distribution of the latent representations.
> In Appendix A ("Analysis of Latent Point Collapse"), we provide a principled explanation of how our method induces the collapse of same-class latent representations into a single point. This condition represents the minimal entropy achievable by the distribution while preserving discriminability between different classes. This is clearly expressed in the limit of small bin sizes for Shannon entropy, as shown in Eq. 5 of the revised manuscript.
> We thus conclude that our method helps finding the minimal possible entropy of the distribution, and thus creates an IB.
>
> **Q4**: *There seem to be discussions of neural collapse and information theory in the literature that are closely related but are not discussed.*
>
> **A4**: We thank the reviewer for pointing out these works that we had previously overlooked. We have now included both of them in our discussion, which has also been enriched with additional recent work on the matrix mutual information ratio and matrix information entropy difference ratio.
>
> In light of the significant effort we have invested in carefully addressing all of the reviewers' comments during the rebuttal process, we kindly ask you to consider revisiting your score. We would deeply value any positive feedback you may offer.

---

### Official Review · Reviewer_mPU4 · 2024-11-04

**Soundness:** 3
**Presentation:** 2
**Contribution:** 2
**Rating:** 6
**Confidence:** 4

**Summary:**

The paper investigates a phenomenon termed latent point collapse by introducing information bottleneck (IB) layers and modifying the loss function in neural networks. Through an information-theoretic lens, the paper derives the loss function and demonstrates its effects. Experimental results indicate that incorporating IB layers enhances the network's robustness to input perturbations and improves reliability in predictions.

**Strengths:**

* The paper is overall clearly written.
* The experiments on the improved robustness when adopting the IB layer are persuasive and interesting.

**Weaknesses:**

The paper would benefit from a more detailed comparison with the existing Neural Collapse (NC) literature. Based on the reviewer's understanding, the main distinctions between this work and NC research lie in two areas:
* Architectural Addition Between Penultimate Layer and Classifier: This paper introduces an extra architectural component between the penultimate layer and the classifier. However, if we interpret the output of this new component as the features used in NC, the framework here still aligns with the NC paradigm. Additionally, the loss function presented in Equation (3) closely resembles the unconstrained feature models commonly employed in NC studies.
* Class Number Exceeding Feature Dimension: This study explores scenarios where the number of classes can exceed the feature dimension, a setup less frequently examined in NC literature. However, recent works have addressed this case, as in [1] and [2]. The authors may want to discuss these works more closely.

[1] Jiang et al; Generalized Neural Collapse for a Large Number of Classes, 2023

[2] Liu et al; Generalizing and decoupling neural collapse via hyperspherical uniformity gap, 2023.

**Questions:**

* The binary encoding structure shown in Figure 1 appears to represent a single class. Could the authors clarify the relationship between the features of different classes within this structure?

* The paper employs the Swish activation function for all experiments. How would the results differ with the standard ReLU activation? The binary encoding won't be possible in this scenario.

* In Tables 1 and 2, the NOPEN architecture seems to represent the default ResNet model without architectural modifications. How would applying the loss function from Equation (3) directly to the true penultimate layer affect the results? Would it still exhibit the benefits of the Information Bottleneck (IB) method?

---

> ### Author Response · Authors · 2024-11-19
> **Rebuttal [1/2]**
>
> We thank the reviewer for thoroughly reading the manuscript. Please find below our responses to all points raised.
>
> **Q1**: *Architectural Addition Between Penultimate Layer and Classifier*
>
> **A1**: The reviewer is absolutely correct that our framework aligns with neural collapse (NC). In fact, in Appendix E ("Neural Collapse"), we demonstrate that all our training occurs predominantly in the terminal phase, where baseline architectures are evaluated during the natural emergence of NC.
> The enhancements provided by our method are therefore *in addition* to those naturally occurring due to NC. Importantly, only our method *induces* the collapse of same-class latent representations into a single point, as shown in Table 1 and explained through principled arguments in Appendix B ("Analysis of Latent Point Collapse").
> This latent point collapse is achieved through strong regularization applied to a linear layer, which prevents the natural increase in the magnitude of latent representations that occurs during optimization. After further research into the literature, we found that this constraint is indeed utilized in the unconstrained feature model in Ref. [1], as highlighted by the reviewer.
> However, we would like to emphasize that the use of this regularization in Ref. [1] serves as a condition to derive a global optimizer and does not address the specific network enhancements that arise only under significant penalty applied to a linear layer. Additionally, in the theoretical study presented in Ref. [1], the network's performance with and without this regularization is comparable, as the regularization is not employed to induce latent point collapse.
> Our usage of this regularization is fundamentally different: it is applied to a linear layer—a novel and necessary condition—to achieve strong regularization, leading to latent point collapse, Lipschitz continuity of the network, and the formation of an information bottleneck.
> Furthermore, our work proposes a novel technical innovation: adding a linear bottleneck layer prior to the classifier. This approach differs significantly from simply adding regularization to the penultimate layer, which, in architectures like ResNet, is often a global average pooling (GAP) layer. Regularizing the GAP layer does not yield the same benefits as applying strong regularization to a linear layer preceding the classifier.
> Thus, we introduce a novel engineering approach by proposing the use of two linear layers before classification instead of one, where the penultimate layer serves as the final bottleneck. Ref. [1] primarily focuses on theoretical analysis and does not explore the practical utility of this type of regularization on a linear bottleneck layer.
> In the revised version of the manuscript, we have thoroughly referenced and acknowledged in multiple sections that this regularization was theoretically postulated in Ref. [1].
>
> **Q2**: *Literature research.*
>
> **A2**: Thank you for pointing out related works that we were not previously aware of. We have expanded our literature review to include these works and have enriched our discussion by referencing additional studies that connect neural collapse to improved transfer learning and explore neural collapse with loss functions beyond cross-entropy.
>
> **Q3**: *The binary encoding structure shown in Figure 1 appears to represent a single class. Could the authors clarify the relationship between the features of different classes within this structure?*
>
> **A3**: The binary encoding structure does not represent a single class but multiple classes. This has been clarified in the revised version of the manuscript. In fact, all elements belonging to a specific class are confined to only one of the peaks. This type of binary differentiation enables the distinction between different classes while achieving the minimum possible entropy associated with the distribution, which creates an information bottleneck. These points have been more explicitly stated in the revised version of the manuscript.
>
> [1] Zhihui Zhu, Tianyu Ding, Jinxin Zhou, Xiao Li, Chong You, Jeremias Sulam, and Qing Qu. A geometric analysis of neural collapse with unconstrained features. NeurIPS (2021)

---

> ### Author Response · Authors · 2024-11-19
> **Rebuttal [2/2]**
>
> **Q4**: *The paper employs the Swish activation function for all experiments. How would the results differ with the standard ReLU activation? The binary encoding won't be possible in this scenario.*
>
> **A4**: In our experiments, we use the Swish activation function in the backbone architecture, as it has been shown to provide better performance compared to other activations such as ReLU. However, we believe our results are independent of the specific activation function employed in the backbone.
> The critical point lies in the activation function used in the layer where we apply L2 regularization. If an activation function such as ReLU is used on that layer, binary encoding will not be possible, as discussed in the manuscript
>
> **Q5**: *In Tables 1 and 2, the NOPEN architecture seems to represent the default ResNet model without architectural modifications. How would applying the loss function from Equation (3) directly to the true penultimate layer affect the results? Would it still exhibit the benefits of the Information Bottleneck (IB) method?*
>
> **A5**: We applied the loss function from Equation (3) directly to the penultimate layer using an increasing scalar factor, as described in Appendix D ("Training and Architecture Details"). However, training did not converge; the network failed to reach the terminal phase where it could correctly classify all, or even a significant portion, of the elements in the training set.
> The reason is that adding the L2 penalty to a non-linear layer constrains all elements to zero, preventing the development of a symmetric structure such as binary encoding. In binary encoding, class points are symmetrically located with respect to the origin, enabling differentiation between classes while minimizing entropy.
>
>
> Given the substantial effort we have dedicated to thoroughly addressing the reviewers' comments during the rebuttal process, we kindly request that you consider the possibility of revising your score. We would greatly appreciate any positive feedback you might provide.

---

> > ### Comment · Reviewer_mPU4 · 2024-11-25
> > **Thanks for the response**
> >
> > I thank the authors for their detailed response and apologize for my delayed reply.
> >
> > I have read the updated manuscript as well as the feedback from other reviewers. Upon re-reading, I realize that I previously misunderstood the demonstration in Figure 1, which may have affected my interpretation of the paper.
> >
> > Having revisited the work, I appreciate the interesting binary coding hypothesis and the empirical benefits highlighted by the authors. However, I feel there is still a lack of empirical support for this hypothesis. In Section 2.2, the authors state:
> >
> > > More precisely, we find that at each node of an IB layer, latent representations can approximately assume one of two values, thereby forming a binary encoding.
> >
> > Please correct me if I’m mistaken, but did the authors provide empirical evidence to demonstrate this?

---

> > > ### Author Response · Authors · 2024-11-25
> > >
> > > We thank the reviewer for their valuable feedback and appreciation of our work.
> > >
> > > **Q**: *Please correct me if I’m mistaken, but did the authors provide empirical evidence to demonstrate that "at each node of an IB layer, latent representations can approximately assume one of two values, thereby forming a binary encoding"?*
> > >
> > > **A**: We sincerely thank the reviewer for highlighting the absence of such empirical evidence in the initial version. We have now addressed this in the revised manuscript. Specifically, in Appendix D, titled "Binarity Hypothesis", we include a table presenting both the average and minimum values for the GMM model scores, as well as the relative peak distances between the two modes of the Gaussian model. The results show that even in the worst-case scenario—represented by the minimum values—the relative distance between the peaks remains substantial, providing robust support for the validity of our binarity hypothesis across all dimensions.
> > >
> > > We hope this revision satisfactorily addresses the reviewer’s concerns. If so, we kindly request that the reviewer consider re-evaluating their score.

---

> > > > ### Comment · Reviewer_mPU4 · 2024-11-25
> > > >
> > > > Thank you to the authors for the updated manuscript. The message is much clearer now.

---

> > > > > ### Author Response · Authors · 2024-11-27
> > > > > **Thank you**
> > > > >
> > > > > Thank you for engaging in this constructive revision process, which has greatly helped us improve our manuscript.

---

### Official Review · Reviewer_qAFB · 2024-11-04

**Soundness:** 2
**Presentation:** 3
**Contribution:** 2
**Rating:** 3
**Confidence:** 4

**Summary:**

The paper proposes a new approach to compress feature vectors by integrating a bottleneck layer and incorporating L2 norm regularization for the newly added layer. This method is grounded as an surrogate for the Lagrangian optimization framework, aiming to minimize mutual information loss.

**Strengths:**

- The proposed approach is straightforward and easy to implement, requiring the addition of only a single bottleneck layer and minimal modification of the loss function
- The empirical results effectively demonstrate the method’s ability to reduce entropy, aligning well with the theoretical intuition presented in the paper.

**Weaknesses:**

- The empirical improvements are unconvincing, as the study lacks comparisons with other established regularization techniques.
- The experimental setup is also limited, with no evaluations conducted on large-scale datasets, such as ImageNet for image classification or Wikitext for language modeling.
- The paper does not provide a theoretical guarantee that the proposed method reduces mutual information loss.

**Questions:**

- Could you include additional baselines to compare your method with other regularization techniques for a more comprehensive evaluation?
- Since your approach appears to induce neural collapse, could you provide a comparison with methods that explicitly leverage neural collapse to enhance model performance?
- Could you offer a theoretical proof or rationale to substantiate that minimizing the proposed loss function effectively reduces mutual information loss?
- Could you consider adding experiments on large-scale datasets to enhance the empirical evaluation of the paper?

---

> ### Author Response · Authors · 2024-11-19
>
> We thank the reviewer for thoroughly reading the manuscript. Please find below our responses to all points raised.
>
> **Q1**: *Could you include additional baselines to compare your method with other regularization techniques for a more comprehensive evaluation?*
>
> **A1**: In the revised version, we added another regularization technique as a baseline: L1 regularization on the penultimate layer, which promotes sparsity in the penultimate layer representations.
> In Table 1, we observe that this regularization induces further clustering of the latent representations, as reflected in the value of $\Sigma_W$ compared to baselines that do not employ this regularization. However, the $\Sigma_W$ values are dramatically different from those of architectures implementing an IB layer. This demonstrates that only our method induces latent point collapse, resulting in the network enhancements we describe.
>
> **Q2**: *Since your approach appears to induce neural collapse, could you provide a comparison with methods that explicitly leverage neural collapse to enhance model performance?*
>
> **A2**: Our method fundamentally differs from neural collapse (NC). NC is characterized by the convergence of same-class latent representations to the vertices of an ETF simplex. In contrast, our method ensures convergence toward the vertices of a hypercube and, more importantly, induces the collapse of same-class latent representations into a single point, which NC does not.
> In Appendix E ("Neural Collapse"), we show that while training predominantly takes place in the terminal phase of training, Table 1 demonstrates that only our method induces single-point collapse. Consequently, the benefits of our method are in addition to those naturally arising from NC.
> Moreover, we are not aware of any method that explicitly leverages NC to enhance both robustness and generalization. Any improvements associated with NC are typically due to its natural development during training rather than being induced by a specific method. If the reviewer is aware of any such method, we kindly request that they let us know.
>
> **Q3**: *Could you offer a theoretical proof or rationale to substantiate that minimizing the proposed loss function effectively reduces mutual information loss?*
>
> **A3**: In Appendix B ("Information Bottleneck in Deterministic DNN Classifiers"), we demonstrate that minimizing the IB Lagrangian optimization problem can be reduced to minimizing the entropy of the probability distribution of the latent representations.
> In Appendix A ("Analysis of Latent Point Collapse"), we provide a principled explanation of how our method induces the collapse of same-class latent representations into a single point. This condition represents the minimal entropy achievable by the distribution while preserving discriminability between different classes. This is clearly expressed in the limit of small bin sizes for Shannon entropy, as shown in Eq. 5 of the revised manuscript.
>
> **Q4**: *Could you consider adding experiments on large-scale datasets to enhance the empirical evaluation of the paper?*
>
> **A4**: As stated in our general rebuttal comment, conducting experiments on large-scale datasets is beyond our current computational resources.
> The datasets we use in this study are already widely employed in many impactful works published at top machine learning conferences. Furthermore, we corroborate our experimental results with principled explanations of the phenomenon we identify.
>
>
> Considering the significant efforts we have invested in addressing the reviewers' comments during the rebuttal stage, we kindly ask whether you might consider revising your score. We would be sincerely grateful for any positive feedback you can provide.

---

### Author Response · Authors · 2024-11-19
**Rebuttal**

Dear Reviewers,

We sincerely thank you for thoroughly reading our manuscript and providing valuable feedback.
We have uploaded a revised version of the manuscript, where we believe the main topic and achievements of this paper are more clearly articulated.
In this paper, we develop a method to induce the collapse of same-class latent representations into a single point. This specific condition renders the network Lipschitz, providing theoretical guarantees for improved robustness. We empirically validate this condition, demonstrating an order-of-magnitude improvement in robustness, which we consider the primary achievement of this work.
The latent point collapse also leads to a reduction in the entropy of the distribution that generates the latent representations. This creates an information bottleneck, which, as noted in the literature, is associated with improved generalization. Consistent with this theory, we observe a small but statistically significant improvement in accuracy. In this revised version, we emphasize in both the abstract and the conclusion that this improvement in generalization is small.
However, in addition to discussing a previously unknown phenomenon occurring in the latent space of deep neural network classifiers—namely latent point collapse—the primary utility of this method lies in its significant improvement in robustness.

In addition, we have included another baseline, as suggested by one of the reviewers. Specifically, we compare the effect of adding L1 regularization to the penultimate layer, demonstrating that the enhancements achieved by our method are distinct from those obtained through other regularization techniques. We notice that we had already compared our method against dropout in the original manuscript.
In the revised version, we compare against only one architecture implementing dropout instead of two, as in the initial submission. We made this change to avoid redundancy and to maintain the clarity and readability of the tables and figures, ensuring that the ablation study does not include an excessive number of baselines.

We believe we have addressed all the points raised by the reviewers, with the exception of the request to conduct tests on larger datasets such as ImageNet.
In our study, we performed an extensive ablation analysis involving 8 different architectures, tested across 5 learning rates, and using 5 different initializations. These experiments were designed to provide an accurate and thorough description of the phenomenon we discuss in the paper.
We estimate that replicating these experiments on a large-scale dataset like ImageNet would require between 300,000 and 500,000 GPU hours, a computational cost we currently cannot accommodate.
We respectfully note that requiring such results may disproportionately disadvantage smaller research groups that lack access to such substantial computational resources.
We would also like to point out that numerous exceptional works published at top conferences such as ICLR have employed the same datasets as we do in this study.
Therefore, we appeal to the reviewers’ understanding to accept our experimental settings and to evaluate our work based on the results we provide, which are supported by sound and principled explanations.

---

> ### Author Response · Authors · 2024-11-19
> **LaTeXDiff file**
>
> To highlight the changes made in the revised manuscript, we have created a LaTeXDiff file that documents all modifications. The file can be viewed using the following link: https://anonymous.4open.science/r/latent_point_collapse-A0B4/diff.pdf

---

> ### Author Response · Authors · 2024-11-25
> **New revision**
>
> Dear Reviewers,
>
> To address additional points raised by some of the reviewers, we have uploaded an updated version of the manuscript. The changes are confined to Appendix B and Appendix D.

---

### Meta-Review · Area_Chair_HJRs · 2024-12-17

**Metareview:**

The paper introduces latent point collapse, a method that encourages within-class latent representations to converge into single points in the latent space of deep neural network classifiers. This collapse creates an information bottleneck that enhances the robustness, generalization, and reliability of the network.

Reviewers generally agree that the method is easy to implement and brings incremental accuracy gains. However, there are major concerns along the following lines:

- Lack of a clear comparison with Neural Collapse. The proposed method conceptually overlaps with Neural Collapse (NC). While the authors provided clarifications, they did not fully resolve reviewers' skepticism. I share the view that the distinction between the within-class collapse induced here and the natural behavior of NC remains unclear.

- Lack of Large-Scale Experiments on datasets like ImageNet. The authors defended this with computational resource constraints, but given that the improvement on smaller datasets CIFAR / SVHN is quite minor, I’d agree with the reviewers that a large scale experiment is necessary to justify the approach.

- Lack of Theoretical Rigor, both for 1) the claim that class points converge to single points or vertices of a hypercube, and 2) the claim that collapsing leads to benefits in robustness / generalization. Regarding the latter, the only argument I found is based on the Lipschitzness as in Eq. (4), but that is still quite a weak argument as it only applies to within-class points but not to between-class points.

Finally, the topic of within-class feature distribution has been extensively discussed in the literature such as Center Loss (https://ydwen.github.io/papers/WenECCV16.pdf) and many followups. There are also works such as Maximal coding rate reduction (https://arxiv.org/abs/2006.08558) that argue otherwise, i.e., encouraging diversity brings robustness benefits. The authors are encouraged to thoroughly review the literature and clarify how their method compares and contributes beyond these existing approaches.

**Additional Comments On Reviewer Discussion:**

Concern: Lack of comparison with Neural Collapse (NC).

Response: The authors clarified distinctions between their method and NC, arguing that latent point collapse forces same-class representations to converge into single points, unlike NC's simplex vertices. Additional explanations were provided in Appendix E.

Concern: Limited theoretical rigor regarding latent point collapse.

Response: The authors added further analysis in Appendix A and Appendix D, showing that latent points approximate vertices of a hypercube. Reviewer gEVm remained unconvinced.

Concern: Marginal improvements in generalization.

Response: The authors emphasized that robustness, rather than accuracy, is the primary benefit. They argued that generalization improvements are statistically significant, albeit small.

Concern: Absence of large-scale experiments.

Response: The authors explained that computational constraints prevented experiments on datasets like ImageNet. They appealed to the reviewers to accept the study's current scope.

Concern: Empirical evidence for binary encoding hypothesis.

Response: The authors added experiments in Appendix D, demonstrating that latent representations assume two distinct values along each dimension.

---

### Decision · Program_Chairs · 2025-01-22

Reject